# A Review of EPDM (Ethylene Propylene Diene Monomer) Rubber-Based Nanocomposites: Properties and Progress

**DOI:** 10.3390/polym16121720

**Published:** 2024-06-17

**Authors:** Naiara Lima Costa, Carlos Toshiyuki Hiranobe, Henrique Pina Cardim, Guilherme Dognani, Juan Camilo Sanchez, Jaime Alberto Jaramillo Carvalho, Giovanni Barrera Torres, Leonardo Lataro Paim, Leandro Ferreira Pinto, Guilherme Pina Cardim, Flávio Camargo Cabrera, Renivaldo José dos Santos, Michael Jones Silva

**Affiliations:** 1School of Engineering and Science (FEC–UNESP), São Paulo State University, Rosana 19274-000, SP, Brazil; naiara.costa@unesp.br (N.L.C.); carlos.hiranobe@unesp.br (C.T.H.); henrique.cardim@unesp.br (H.P.C.); leonardo.paim@unesp.br (L.L.P.); leandro.f.pinto@unesp.br (L.F.P.); guilherme.cardim@unesp.br (G.P.C.); renivaldo.santos@unesp.br (R.J.d.S.); 2School of Technology and Sciences (FCT–UNESP), São Paulo State University, Presidente Prudente 19060-900, SP, Brazil; dognanig@gmail.com; 3Mechanical Engineering Department, Pascual Bravo University Institution (IUPB), Medellín 050036, Colombia; juan.sancjezg@pascualbravo.edu.co (J.C.S.); jaime.jaramillo@pascualbravo.edu.co (J.A.J.C.); 4Industrial Design Engineering Department, Arts and Humanities Faculty, Metropolitan Institute of Technology (ITM), Medellín 050036, Colombia; giovannibarrera@itm.edu.co

**Keywords:** EPDM, synthetic rubber, nanocomposites, nanofillers, properties

## Abstract

Ethylene propylene diene monomer (EPDM) is a synthetic rubber widely used in industry and commerce due to its high thermal and chemical resistance. Nanotechnology has enabled the incorporation of nanomaterials into polymeric matrixes that maintain their flexibility and conformation, allowing them to achieve properties previously unattainable, such as improved tensile and chemical resistance. In this work, we summarize the influence of different nanostructures on the mechanical, thermal, and electrical properties of EPDM-based materials to keep up with current research and support future research into synthetic rubber nanocomposites.

## 1. Introduction

The search for versatile, lightweight materials with suitable hardness, a large area, appropriate coercivity, low production cost, and the ability to modify their chemical and physical properties when subjected to external stimuli has grown in recent decades [1,2]. With the recent developments in nanotechnology and nanomaterials, producing materials with excellent properties that meet their designed requirements has become possible. Among these classes of materials, polymer-based nanocomposites, especially those based on rubber matrices, such as ethylene propylene diene monomers (EPDM) [3,4], styrene–butadiene–styrene (SBS) [5,6], and natural rubber (NR) [7,8,9], are significant.

EPDM is one of the most studied and used synthetic rubbers by the scientific community and industry [10]. EPDM rubber is a member of the unsaturated polyolefin family; it is prepared via the polymerization of propylene and ethylene with a small quantity of non-conjugated diene (approximately 3–9%) [11,12,13,14]. Figure 1 illustrates the chemical structure of EPDM, which contains a saturated hydrocarbon backbone.

In addition to its excellent heat resistance, elasticity and flexibility at low temperatures, weathering properties, oxidation resistance, ozone resistance, and aging resistance, EPDM has also been applied to a wide variety of applications, including weather-stripping for automobiles, roofing membranes, sealants, tubing, belts, radiators, thermal insulation, and electrical insulations [11,15,16,17,18,19,20].

Typically, the properties of EPDM rubber are improved through vulcanization by creating a crosslinked structure using sulfur as the vulcanizing agent and chemical accelerators [21]. However, nanofiller-based reinforcement has also been used as an alternative method for improving the physical and chemical properties of the EPDM matrix as well as reducing its production cost [22,23]. One advantage of EPDM compared to synthetic rubber and NR is its capacity to accept large amounts of nanofillers, which can significantly improve its properties [24]. Various nanofillers with inherent properties have been used to reinforce EPDM rubber and obtain nanocomposites with improved final properties. Among these nanofillers, organic-based nanoparticles, such as carbon nanotubes (CNTs) [25,26], multi-walled carbon nanotubes (MWCNTs) [27,28], graphene [29,30], and carbon black (CB) [31,32,33], and inorganic-based nanoparticles, such as nanoclay [34,35,36], nanosilica [37,38], montmorillonite [11,39], and polyhedral oligomeric silsesquioxane (POSS) [40,41], should be highlighted.

The incorporation of different nanofillers into the EPDM matrix can expand the application of EPDM in various industrial and technological areas, for example, EMI (electromagnetic interference), shielding effectiveness in electric devices [29,42,43], sensors [44,45], nuclear applications [46], thermal and electrical insulating materials [11], and solid rocket motor insulation [38,47], among others.

These diverse applications are due to the improved final mechanical and morphological properties caused by the dispersion of different nanofiller types in the polymeric matrices. For example, the dispersion of nanofillers in EPDM-based nanocomposites can improve the thermal stability and conductivity of the nanocomposites, allowing for application in the field of thermal energy storage and thermal management of electronic devices [48]. George et al. [38] evaluated the insulation behavior of nanosilica-filled EPDM/Kevlar fiber and established that a 220% improvement in char residue was observed with enhanced thermal stability and mechanical properties. The authors evaluated the insulation behavior of nanosilica-filled EPDM/Kevlar fiber and established that a 220% improvement in char residue was observed with enhanced thermal stability and mechanical properties [38]. These properties are attributed to the load of Kevlar in the composites, which maintains thermal stability and increases the char residue. However, nanosilica does not significantly improve the composite [38]. Guo et al. [49] studied the effects of the incorporation of MWCNTs on the char residue and carbothermal reduction reaction in EPDM-based nanocomposites. They observed that the MWCNT network structure in the char has a directional effect on in situ SiC (silicon carbide) formation, which improves the ablation resistance of the nanocomposite. Zhang et al. [50] investigated the impact of adding flame-retardant and dendrimer-modified organic montmorillonite (FR-DOMt) on the thermal and mechanical properties of EPDM-based nanocomposites. It was observed that an increase in the quantity of FR-DOMt in the nanocomposite enhanced the thermal stability and flame retardance of the EPDM matrix. Lu et al. [44] successfully prepared a highly stretchable and sensitive sensor based on a graphene nanoplatelet (GnPs)/EPDM nanocomposite with excellent heat dissipation performance. Owing to the good dispersion of the GnPs in the EPDM-based nanocomposite, the nanocomposite-based sensor had a low percolation threshold value when using approximately 2.9 wt.% GnPs and a thermal conductivity of 0.72 W/m K when using 7 wt.% GnPs. The results indicate that the GnPs/EPDM sensor shows excellent potential for monitoring deformation and motion in the human body [44].

Several types of nanofillers have been incorporated into EPDM for reinforcement purposes; one of the most important parameters used to characterize reinforcement materials is their specific surface area, which should be in the order of hundreds of square meters per gram, as it is directly related to the size of the material particles [51]. Another critical parameter used to describe these reinforcement materials is the relationship between the average length and the diameter, known as the aspect ratio [52]. In addition to the parameters above, the mechanical properties of the nanocomposites are improved through the good dispersion of the nanofillers, which can be achieved through different forms of processing [52].

The dispersion of nanofillers is intricately linked to three conditions. The first is related to nanofiller surface chemistry (presence or absence of active functional groups), which can compromise the compatibility between the filler and the polymeric matrix. The second is the interfacial interaction of the fillers with the polymeric matrix, which can be chemical or physical and depends on the formation of aggregates due to filler–filler interactions. The third condition is the correct choice of a polymeric matrix; there is a need for compatibility between the type of nanofiller and the polymeric matrix.

The parameters should be balanced to achieve the desired reinforcement properties in the nanocomposites. For example, when the nanofiller concentrations are relatively high, the quality of reinforcement and interactions between the matrix and the nanofillers can deteriorate owing to the smaller interfacial contact area [53]. Consequently, aggregates are formed that induce various localized stress concentrations, which leads to the initialization of defects, such as crack propagation and failure [54]. The relatively increased hardness and tension can improve the mechanical properties due to more nanofillers, but the abrasion resistance is reduced owing to concentrated stress points [55]. In such cases, it is necessary to improve the surface of the load with various treatments, for example, chemical treatments, so that the interfacial load–matrix interaction is optimized [56]. This causes multiple properties to improve, possibly reducing the volume of the incorporated load [56].

EPDM is an elastomer that exhibits acceptable and valuable mechanical and dielectric properties and shows specific resistance to several conditions, such as oxidation, chemical attack, and resistance to weather effects [57]. A modification using tungsten oxide nanoparticles was made by Sang et al. [58] and incorporated into EPDM foam to improve thermal storage capacity and seawater resistance; such properties could be potential applications in marine sportswear products, among other fields where these properties are desirable. In another study, Bianchi et al. [59] obtained EPDM foams that were filled with different amounts of paraffin, a typical phase change material (PCM) with a melting temperature of approximately 70 °C, to develop rubber foams with thermal energy storage (TES) capabilities [59]. This system utilizes a heat absorption/release principle, using materials called PCMs, which are excellent thermal conductors due to their high energy storage capacity [57,58,60,61,62]. The main characteristic of these materials is that they can be used for thermal insulation and heating and cooling purposes, especially in the field of Heating, Ventilation, and Air Conditioning (HVAC) [57,62,63]. Additionally, EPDM/paraffin compounds have been reported to be helpful in thermal energy storage applications in buildings [64,65]. As a positive consequence of materials with TES properties being used in construction, TES technology helps to balance the energy demands of buildings by reducing energy peaks for air conditioning processes [66,67].

EPDM rubbers can be classified as amorphous polymers, and the addition of reinforcing fillers is considered a significant factor in improving the mechanical properties and strength of rubber regarding unfilled EPDM rubber [68]. An essential application in industrial electric fields is related to the use of fillers and nanofillers in EPDM rubbers, which are used to suppress space charge in high-voltage direct current (HVDC) cables, which is a factor that promotes insulation breakdown. Yang et al. discussed using various types of nanofillers [69]. The first type of fillers includes inorganic nanofillers, such as ZnO and zeolite, inorganic nano-carbon, oxides such as SiO_2_, and organic chemical modification, such as crosslinked polyethylene/organic nano-montmorillonite (XLPE/O-MMT), another polymer that exhibits a positive response in the presence of low electrical fields relative to suppressing space charge accumulation [70]. On the other hand, carbon series of nanofillers, such as graphene and graphene oxide (GO), including a carbon nanotube, exhibit a positive effect on space charge suppression, suggesting their use as potential filler materials for (HVDC) cables. Another critical problem associated with high-voltage power cables is related to humid environments, which can promote a water tree phenomenon responsible for the degradation of dielectric properties, consequently reducing the lifespan of XLPE when improving the properties of these cables [71]. Qingyue et al. [72] used three kinds of dopants, nano-montmorillonite (MMT), spherical nano-silicon dioxide (SiO_2_), and polar ethylene-vinyl acetate copolymer (EVA), over XPLE samples using a melt blending process to analyze several properties. The results showed that final composites exhibit a better elastic modulus, better toughening, and the lowest growth of the water tree effect compared with XLPE without dopants [72].

Blends of EPDM/styrene butadiene rubber (SBR) have been analyzed by some authors [73,74,75] regarding the effect of different nanocomposite blends, such as reinforcing agents, and the consequent impact on mechanical properties. Broad engineering applications can be found using these kinds of (EPDM/SBR) rubber compounds, such as tires, seals, conveyor belts, and electrical cables, among others. The reinforced composite is essential for this (EPDM/SBR) composite. The addition of nanoclay (NC) and nanosilica (NS) can promote better results regarding tensile strength, tear strength, hardens, and abrasion resistance [73]. On the other hand, Badahar and Zawawi [75] used single-walled carbon nanotubes (SWCNTs) to reinforce EPDM/SBR nanocomposites, and the results showed values close to a 20% improvement in harder, better rheology, and viscoelastic response as well as thermomechanical performance, which are essential characteristics for improving shock absorption engineering applications. Deng et al. [76] designed and analyzed a double foaming system composed of EPDM/SBR and thermoplastic rubber/TPR composite foam to improve the mechanical properties of this polymeric foam, particularly in tear strength, which is considered an essential characteristic in terms of reducing costs and achieving greater toughness, with potential applications in automotive, aerospace, and construction. The authors suggest that the bimodal structure of the foam cells and the reduction in the crystallization of molecular chains are critical factors in achieving higher tear strength [76].

Given the importance of EPDM as a polymeric matrix and its wide application in industry and technological developments, this manuscript aims to present a brief review of the thermal, mechanical, electrical, morphological, and rheological properties of EPDM-based nanocomposites modified with nanofillers. Overall, this review emphasizes the role of nanofillers in improving EPDM-based nanocomposites across multiple dimensions, including mechanical, thermal, electrical, morphological, and rheological properties. As a result of these studies, a broader understanding of EPDM-based nanocomposite properties and their applications is provided.

## 2. Mechanical Properties

This section discusses improvements in the mechanical properties of EPDM-based nanocomposites incorporating nanofillers, including tensile strength, hardness, and abrasion resistance.

Kermani et al. [77] produced hybrid nanocomposites based on EPDM–butadiene rubber (XSBR) mixed with different concentrations of MWCNTs, using maleic anhydride grafted on EPDM rubber (EPDM-g-MA) as a compatibilizer. The effect of the MWCNT concentrations on the mechanical properties of the hybrid nanocomposites was studied. The authors noted that EPDM-g-MAH improved the distribution of the MWCNTs within the polymer matrix, resulting in a uniform distribution of MWCNTs with a small amount of aggregation. In addition, the mechanical properties, such as modulus, tensile strength, hardness, and elongation at break, of the compatible EPDM/XSBR nanocomposite were better than those of incompatible composites. Vayyaprontavida Kaliyathan et al. [78] showed that rubber blends are pivotal in tailoring the specific properties sought in rubber-based products. The performance of these blends is significantly influenced by their underlying morphology [78]. Detailed examination of this morphology is made possible by applying various microscopy techniques, allowing for a deeper understanding of their structure and behavior. Notably, achieving miscibility in rubber blends is a rarity; however, additives can be introduced to enhance compatibility between different rubber types. This approach serves as a method to bridge the gap in terms of properties and create more harmonious blends despite the inherent challenges of achieving complete miscibility in these mixtures [78]. As a result, Vayyaprontavida Kaliyathan et al. [78] analyzed the mechanical properties of rubber–rubber blends. A notable enhancement was observed in the mechanical characteristics of these blends throughout the blending process. Employing specific mixing equipment such as internal mixers and extruders proves instrumental in refining the formation of these blends, ensuring a more uniform and optimized composition [78]. Addressing incompatibility issues within blends and incorporating compatibilizers emerge as a solution to render these initially incompatible blends more harmonious. This addition bridges the gap between disparate rubber types, facilitating a more cohesive and compelling blend.

According to Sowińska et al. [79], Ionic Liquids (ILs) stand out in elastomer technology owing to their distinct and advantageous properties. Supported Ionic Liquid-Phase (SILP) materials serve to anchor ILs onto solid supports, a technique explored in this study to understand its impact on the properties of EPDM elastomers [79]. The investigation delves explicitly into how SILPs exert control over the vulcanization process without causing crosslink density or thermal stability alterations. SILP materials play a pivotal role in regulating the vulcanization process of EPDM without causing any degradation in crosslink density [79]. Moreover, they contribute positively by enhancing the tensile strength and hardness of the resulting vulcanizates. These modifications signify an improvement in the mechanical properties of the EPDM material facilitated by SILP incorporation. EPDM vulcanizates exhibit robust resistance to thermo-oxidative aging when subjected to a temperature of 100 °C. This underscores the material’s durability and stability under harsh environmental conditions [79]. Furthermore, the analysis conducted through Dynamic Mechanical Analysis (DMA) sheds light on the influence exerted by SILPs on the mechanical loss factor, also known as tan delta (tan δ), demonstrating their impact on the viscoelastic behavior of EPDM [79]. The findings from this study [79] highlight that while employing SILPs, marginal enhancements are noted in the tensile strength and hardness of the resultant vulcanizates, suggesting a potential for SILPs to exert a subtle yet discernible influence on the mechanical characteristics of EPDM elastomers [79].

Vishvanathperumal and Anand [80] prepared EPDM/SBR hybrid composites reinforced with nanoclay (NC) and nanosilica (NS) and investigated the synergistic effect of NC and NS on the mechanical properties of the EPDM/SBR hybrid nanocomposites. Three different crosslinking systems were used: dicumyl peroxide, sulfur, and a combination of peroxide and sulfur. The best tensile strength was achieved using 4 phr of NC; using higher mass concentrations reduced the tensile strength [80]. This behavior was attributed to the adsorption of the MBTS and TMTD accelerator on the silica surface by the silanol groups. The authors found that sufficient Si-69 in filler reduces the adsorption process. In addition, the high NS content exhibits strong NS–NS interactions, compromising abrasion resistance, among other properties [80]. The concentrations of NS (4 phr) and NC (7.5 phr) in the sulfur-cured EPDM/SBR hybrid nanocomposites play an essential role in the microstructural and mechanical properties of the nanocomposites. The study revealed that nanocomposites containing 7.5 phr of NC and NS exhibited the best mechanical and abrasion resistance characteristics [80].

Within Rostami-Tapeh-Esmaeil et al. [81], comprehensive insights are offered into a spectrum of aspects concerning rubber foams. This includes detailed information encompassing formulation strategies, curing methodologies, diverse production techniques, an intricate study of morphology, an extensive examination of properties, and an exploration of the wide-ranging applications these rubber foams find across different industries [81]. Rubber foams exhibit superior attributes such as increased flexibility, abrasion resistance, and enhanced capabilities for energy absorption [81]. Formulation choices and various processing parameters significantly influence the physical and mechanical properties of these foams. Tensile testing is infrequently employed with foams primarily due to challenges in securely gripping the material for analysis. Instead, compressive loading finds more prevalence, particularly in applications involving cushioning and packaging [81]. The mechanical behavior of foams is significantly influenced by their morphology and the thickness of their cell walls [81]. In particular, an increase in the dispersion of cell wall thickness is observed to correspond with reductions in both Young’s modulus and shear modulus. Introducing fillers such as rice husk powder and kenaf particles effectively enhances the tensile strength and stiffness of foams [81]. Moreover, the compression stress experienced by foams tends to be higher in instances where more giant cells and a higher content of foaming agents are present. Additionally, a higher content of foaming agents correlates with increased compression set observed in foams, indicating a greater degree of deformation or relaxation under sustained compressive loads [81].

To enhance the properties of an NR/EPDM blend, Lee et al. [82] introduced a phlogopite mineral filler into the mixture. This process involves the incorporation of compatibilizers, specifically Amino Ethyl Amino Propyltrimethoxy Silane (AEAPS) and Stearic Acid (SA). The resulting NR/EPDM/phlogopite/AEAPS composite displayed comparable properties to those observed in the NR/EPDM/CB composite, suggesting similar performance between the two materials [82]. This composite of NR/EPDM/phlogopite/AEAPS holds the potential to serve as a viable alternative to costly fillers like CB, offering similar properties and performance while potentially reducing production costs [82].

A novel approach has been introduced to enhance the toughness of polypropylene (PP) by incorporating nanostructured rubber into the material by Chang et al. [83]. This involves a scalable method for producing PP nanocomposites, leveraging crosslinked EPDM nanofibrils to bolster toughness. As EPDM loading increases, the yield strength and elastic modulus of the PP nanocomposites consistently decline. This trend suggests the incorporation of relatively softer inclusions into the matrix. In conventional rubber-toughened systems utilizing rubber microparticles, a notable decline is observed in both yield strength and modulus. This decrease surpasses what the Rule of Mixtures theoretically predicts. The measurement of the viscoelastic moduli (*G*′, *G*″) and complex viscosity (*η*) in both PP nanocomposites and fibrillar composites illustrates the distinct contributions of individual polymeric constituents (PP and EPDM) to the overall material response [83].

Azizli et al. [84] produced several elastomeric nanocomposites by mixing NR with EPDM compatible with maleic anhydride (EPDM-g-MA) reinforced with different amounts of GO (1.0, 3.0, 5.0, 7.0, and 10.0 phr); the mechanical properties were evaluated theoretically and experimentally. The results showed that incorporating GO improved the nanocomposites’ mechanical properties, such as hardness, tensile strength, elongation at break, and modulus [84]. These improvements are due to variations in the NR content that increase the crosslinking points, the size of the surface area, and the dispersion of GO in the nanocomposites [84].

A study by Burgaz and Goksuzoglu [85] synthesized isotropic Magnetorheological Elastomers (MREs) with potential use in automotive applications. Among the formulations tested, EPDM/CB/Carbonyl Iron Powder (CIP) MREs demonstrated superior properties compared to EPDM/CB/Bare Iron Powder (BIP) MREs. The particles known as CIP displayed distinct characteristics in contrast to BIP particles within these MRE formulations. CIP particles exhibited lower damping factor, Payne effect, elastic modulus, and hardness but demonstrated higher values in tensile strength and elongation at break. Moreover, the sample comprising 5 phr CIP and 60 phr CB showcased a substantial MR effect of 77%, signifying the material’s responsiveness to a magnetic field [85].

Several reported studies demonstrate nanofiller dispersion’s effect on the mechanical properties of EPDM-based nanocomposites. Nazir et al. [86] tested the effect of boron nitride (BN) on the breakdown, surface tracking, and mechanical performance of EPDM for high-voltage insulation. It was established that by obtaining a hybrid composite formed using BN-based micro and nanoparticles, better tensile strength (*σ_t_*) and elongation at break (*ε_b_*) values were achieved; for example, the EPDM matrix filled with BN micro-25 wt.% + nano-5 wt.% particles presented better mechanical properties (*σ_t_* = 5.2 MPa and *ε_b_* = 353%) than EPDM filled with BN micro-30 wt.% (*σ_t_* = 4.06 MPa and *ε_b_* = 302%) [86]. In rubber nanocomposites, the dispersion of nanofillers and the interfacial interactions between the rubber matrix and nanofillers have a significant impact on their properties. The limits are associated with the interaction of the filler with the polymeric phase, as specified by the author [86], and a combination of fillers with sizes on the order of micro and nano, for example, to EPDM-based nanocomposite with 25 wt.% microfillers and 5 wt.% nanofillers exhibited tensile strength of more than 5 MPa and elongation of more than 300%, but not for other combinations or individual load applications. It is also possible to use percolation theory to describe the mechanical behavior of composite materials [9]. This is because when nanofillers are dispersed in a polymeric matrix, they can form a three-dimensional network of geometric contact between them in the composite when their concentration is equal to the mechanical percolation threshold of the nanocomposite [9]. Consequently, when the nanofiller concentration is equal to or greater than the percolation threshold and mechanical tension is applied to the composite, part of the mechanical stress is efficiently transferred from the matrix to the nanofillers [9]. Consequently, when the concentration of nanofiller in the composite approaches the percolation threshold, the mechanical resistance and elastic modulus of the composite tend to increase. This behavior may be related to a significant reduction in the inter-particle distances and an increase in the total surface area of the particles in the co-filled composites [86]. Molavi et al. [87] evaluated the effect of MWCNTs on the mechanical and rheological properties of silane-modified EPDM rubber (VTMSg–EPDM). Even at deficient MWCNT concentrations, enhanced physical properties were observed for the VTMSg–EPDM matrix owing to the grafting reaction that improved the dispersion of the MWCNTs [87]. Nanocomposites with 1.5 wt.% MWCNTs exhibited remarkable elastic modulus and tensile strength improvements. Increasing the MWCNT content also positively affected the storage modulus (*G*′) and complex viscosity (*η**), both of which were enhanced; however, the difference in the *G*′ and *η** values decreased as the frequency increased [87]. As a result of the reduced interparticle distance, the total surface area of the particles increases, improving the processability of the material. In addition to the excellent dispersion and low nanofiller–nanofiller interaction, the ease of processing and incorporation of nanofillers into polymeric matrix increases the nanofiller–matrix interaction. This good dispersion resulted in improved mechanical and thermal properties for the nanocomposite.

## 3. Rheological Properties

The curing characteristics of EPDM-based nanocomposites filled with nanoparticles are established during the vulcanization process. They can be measured using the materials’ rheometric parameters [13,84,88], including the minimum (*ML*) and maximum torque (*MH*), scorch time (*ts*_2_), curing time (*t*_90_), and cure rate index (CRI), which will be discussed in this section.

Khalaf et al. [22] produced a hybrid EPDM/chitin/nanoclay bionanocomposite using a binary loading system formed by nanoclay and chitin from shrimp shells. The purpose of this study was to reuse the residue from previously treated shrimp shells and to evaluate the rheological properties of the bionanocomposite with the addition of nanoclay. In addition, the physical–mechanical properties, swelling parameters, morphological analysis, and water absorption of the vulcanized polymers were studied. Sulfur was used as the crosslinking agent in the EPDM rubber containing 55% ethylene. The curing characteristics were measured using a Monsanto oscillating disc rheometer (ODR -100 s) at 152 °C according to ASTM D-2084-07 [22,89]. Different concentrations of nanoclay (3, 5, 7, and 10 phr) were incorporated into the EPDM/chitin mixtures (5 and 10 phr). The results showed that the *ML* increased with the nanoclay quantity. An increase influenced the viscosity of the EPDM-based nanocomposite; the decrease in fluidity was associated with processing [22]. By contrast, the *MH* value showed a contradictory trend as a decrease in the *MH* values with increased nanoclay load concentration was observed. The authors established that as the amount of nanoclay in the prepared vulcanized polymers increased, there was a reduction in t_90_ and an increase in the ts_2_ and CRI values. This behavior was attributed to the polarity of the silicate layers of the nanoclay, which contributed to the formation of hydrogen bonds and accelerated the curing process at low nanofiller concentrations [22]. Therefore, the authors concluded that the ideal binary charge concentration for EPDM/chitin/nanoclay was 5 and 10 phr of chitin with 3 phr of nanoclay for EPDM rubber applications in industrial heat exchange processes. The different geometries, particle sizes, and surface activities of nanofiller influence the *ML*, which increases with an increase in nanoclay content on hybrid EPDM/chitin/nanoclay composites [22]. The increase in pre-curing time is a result of an increase in the specificity of the rubber compound and the decrease in fluid capacity associated with the processing and dispersion of the load. Contrary to the *ML*, the *MH* decreased with increasing nanoclay quantities and possible interactions with EPDM matrixes [22]. As the nanofiller content in the prepared vulcanizates increases, the curing speed index and optimal curing time decrease [22]. There is a possibility that this is due to the polarity of the clay silicate layers, which facilitate the formation of hydrogen bonds and thus accelerate the curing process at low nanoclay concentrations [22]. In addition, it is possible to reduce waste and promote sustainability by reusing materials such as chitin, from shrimp shells as nanofillers in EPDM. Incorporating recycled materials into EPDM-based nanocomposite can enhance their biodegradability, contributing to the preservation of the environment. Using recycled chitin as a nanofiller in EPDM-based nanocomposite allows for a more comprehensive lifecycle assessment of the rubber products, considering their environmental impact from the point of production to the end of disposal [22]. EPDM-based nanocomposites containing recycled chitin fillers may have improved durability and longevity, potentially extending their useful life and reducing their environmental impact [22]. The manufacturing process can be made more cost-effective using recycled materials by repurposing waste materials, such as chitin from shrimp shells. Using sustainable and cost-efficient production practices, EPDM-based nanocomposite containing recycled chitin fillers may provide a more economical alternative to traditional rubber formulations [22].

Sowińska et al. utilized a rotorless curemeter to analyze the rheological properties of rubber compounds [79]. The samples underwent vulcanization at 150 °C employing *t*_95_ and *t*_80_ (at the point when the torque increases by 95% and 80% of the maximum value, respectively) as the vulcanization times. This study aimed to determine the impact of SILPs on the curing characteristics of EPDM compounds. Figure 2 displays the rheometric curves representing the EPDM compounds’ behavior during the process.

As per the rheological analysis of the developed EPDM biocomposites by Chen et al. [90], both EPDM/WF and EPDM/Si-WF composite samples exhibited typical shear thinning behavior and reduced viscosity as the shear rate increased. The higher WF/Si-WF loadings increased the viscosity due to the filler-induced inhibition of molecular movement. Comparing EPDM/WF with EPDM/Si-WF samples, the former showed loosely stacked viscosities across different WF loading levels, notably with the 50 WF sample displaying significantly higher viscosity [90]. Conversely, EPDM composites with Si-WF displayed closely aligned viscosities across varying filler levels, attributed to improved interfacial compatibility through silane modification. This enhancement resulted in minimal viscosity variation among different filler levels. In the case of samples with DCP, their viscosity curves surpassed that of pure EPDM. The increased viscosity in Si-WF + DCP-based composites is anticipated to stem from crosslinking or molecular entanglement initiated by DCP [90].

Abdelsalam et al. [91] focused on the rheological parameters of *ML*, *MH*, *tc*_90_, and scorch time (*ts*_2_). *ML* gauges the stiffness and viscosity of an unvulcanized elastomer, signifying its processability, observed at the lowest point on the torque–time plot. *MH* represents the peak torque during curing, which is directly linked to the compound’s modulus, reflecting its stiffness. The difference between *MH* and *ML*, Δ*M*, typically estimates the crosslink density. The *tc*_90_ marks the duration to reach 90% of *MH*, while *ts*_2_ signals the vulcanization initiation as torque increases by two units [91]. The rise in *ML*, *MH*, and Δ*M* with escalating CB content reveals enhanced viscosity in the NR/SBR/NBR matrix, suggesting improved interactions between CB and the polymer matrix. *S*_1_ (without filler) displays the lowest *ML*, indicating easier processing than CB-loaded samples [91]. Higher *MH* in filled rubber blends indicates robust interactions between the nanofiller and the polymer matrix, which is attributed to the smaller particle size promoting increased rubber-filler interactions and more excellent resistance to molecular motion, leading to elevated torque values. The rise in Δ*M* relates to increased crosslink density [91]. Moreover, as filler loading increases, both *ts*_5_) and *tc*_90_ decrease [91]. Lower *ts*_5_ signifies better processing ability associated with the heat history that the rubber can withstand before reaching a crosslinked state. This potentially increases the CRI, showcasing the application of CB as a reinforcement and an effective accelerator of NR/SBR/NBR vulcanization, reducing process duration in tire manufacturing. The decline in ts_2_ is attributed to increased crosslinking, while the reduction in tc_90_ is linked to heightened energy input and heat build up due to increased viscosity during mixing. The curing rate index elevation with filler addition stems from CB’s basicity, expediting the vulcanization reaction [91].

## 4. Swelling Properties

The blending of the two materials has been a focus of interest for specific applications in several studies for more than 40 years, including the performance of polymer systems and their variations in elasticity and viscosity at different strain rates and temperatures [92,93,94,95]. An example of a polymer blend is EPDM, which is used in industrial applications owing to its high resistance to aging, heat, polar solvents, and other chemicals and its outstanding electrical characteristics and mechanical properties [96]. There has been considerable interest in the swelling properties of this type of material in several research studies [97,98,99,100,101]. In most cases, this property is associated with the absorption of liquid, which improves the material so that it can exhibit changes in dimension up to 100% of its initial volume [102].

Vishvanathperumal and Anand [103] showed that several parameters affect the swelling properties of reinforced EPDM with nanosilica, such as the matrix shape, type and geometry of filler, the reaction of the solvent on the matrix, temperature, and penetrant, among others, in addition, they identified that crosslinks restrict the swelling-induced extensibility of rubber polymeric chains by reducing the dispersion of the solvent on the matrix. Stelescu et al. [96] also identified parameters associated with swelling: reinforcement density, crosslink density, and solvent absorption. Additionally, it has been shown that the amount of oil used to improve the mixing and processing while maintaining the stiffness influences the swellability of the compounds due to the oil particles that impact the crosslinking process, affecting the stiffness and leading to softness [104].

On the other hand, Abdelsalam et al. showed that swelling could be improved by using a silane-like coupling agent when Al_2_O_3_ is added to the EPDM [105]. Likewise, it has been demonstrated that the swelling resistance is possibly improved via the addition of nanocomposites as nGO with amounts around 6 phr [106]. Colom et al. confirmed the inverse relationship between swelling and crosslink density by showing that increasing this last property reduces the possibility of toluene entering the tighter polymeric network, reducing swelling [107]. A similar phenomenon was demonstrated by Simet et al. when swelling samples within xylene over 24 h, achieving up to 30% variation in the swelling ratio, which is associated with the degradation of the structure due to interactions with the solvent [108]. To minimize dimensional changes, it is possible to use pre-swelling, as carried out by Nijibah et al., which significantly reduces the swelling or shrinking of the membranes when compared to the previous process [109].

## 5. Thermal Properties

Thermal properties refer to the characteristics of a material that describe its behavior in response to changes in temperature. The influence of different types of filler on the thermal behavior of EPDM-based nanocomposites has attracted the interest of researchers since the dispersion of nanoparticles can attribute new properties to the final composite. In this sense, properties such as thermal stability, thermal conductivity, melting point, residue, and glass transition temperature (*T_g_*) can be studied. These analyses include differential scanning calorimetry (DSC), thermogravimetric analysis (TGA/DTGA), dynamic mechanical thermal analysis (DMTA), and thermal conductivity. This section focuses on studying the thermal properties of EPDM-based nanocomposites using some of the main thermal analysis techniques.

### 5.1. TGA Analysis

Thermogravimetric (TGA/DTGA) analysis evaluates the mass of a sample as a function of temperature using a controlled atmosphere, such as inert or oxidizing gas [110]. Using this technique, changes in the mass (loss or gain) and decomposition events of materials, including those based on EPDM, can be analyzed, and the effect of nanofiller dispersions on the thermal stability of the polymeric matrix can be evaluated [110,111].

Several studies have used TGA/DTGA analysis to evaluate the effect of nanofiller dispersions on EPDM-based nanocomposites, such as carbon-based materials (CB, non-functionalized and functionalized carbon nanotubes (fCNT), graphene, etc.) and inorganic-based particles (nanosilica, nanoclay, tungsten bronze nanorods, tungsten oxide, etc.), among others.

Introducing inorganic-based nanoparticles into EPDM-based nanocomposites influences the thermal profile and stability. Sang et al. [58] fabricated a bio-EPDM/tungsten oxide nanocomposite foam with improved thermal storage and seawater resistance and evaluated its thermal properties using TGA/DTGA analysis. Figure 3a shows the thermal profile of a bio-EPDM-based nanocomposite containing various mass ratios of TBNR (tungsten bronze nanorods). Two mass loss events are observed in the 25–700 °C temperature range. The first corresponds to a mass loss of 2% due to the decomposition of the hydrocarbon chain, and a second mass loss event occurs between 430 and 590 °C due to the decomposition of the EPDM polymer. Figure 3b indicates that the maximum decomposition rate temperature (*T_m_*) of the bio-EPDM foams shifts to a higher temperature with an increase in the amount of TBNRs. According to the authors, this behavior indicates that the interfacial interaction between the chain on the surface of the TBNRs and the bio-EPDM matrix enhanced the chemical stability of the bio-EPDM matrix and improved the thermal stability of the nanocomposite [58].

Rana et al. [112] observed similar behaviors for EPDM-based nanocomposites filled with varying amounts of nanoclay using TGA analysis. According to the authors, the mass loss in the main event (between 300 and 500 °C) is lower for neat nanoclay than for the nanocomposites containing 6, 4, and 2 wt.% nanoclay. In contrast, EPDM presented a higher mass loss between 30 and 690 °C. A shift in the *T_m_* peak (DTG curve) to a higher temperature is observed with an increase in the nanoclay quantity relating to neat EPDM. This shift was attributed to the increased thermal stability of the EPDM-based nanocomposite with the increased amount of nanoclay [112]. Zhang et al. [113] also used TG/DTG analysis to evaluate the behavior of an EPDM-based nanocomposite with a crosslinked interfacial design containing nanoclay. The results obtained demonstrate that the incorporation of nanoclay did not influence the thermal profile of the EPDM-based nanocomposite; however, the residue percentage increased significantly with increasing nanoclay content, indicating that the nanoclay content influences the thermal stability of the EPDM matrix.

The influence of organic-based nanoparticles on the thermal properties of EPDM-based nanocomposites has also been investigated in several studies. To assess the dispersion effect of MWCNTs on the physicochemical properties of EPDM insulation for solid rocket motors, Guo et al. [114] used TG/DTG analysis to evaluate the thermal stability and ablation characteristics of two formulations of EPDM (with and without MWCNTs). The study indicated that adding MWCNTs increased the residue rate after thermal decomposition and suppressed the consumption reaction of the char layer under high-temperature conditions [114]. Guo et al. [49] studied the effects of the incorporation of MWCNTs on the char residue and carbothermal reduction reaction in EPDM-based nanocomposites. They observed that the MWCNT could significantly enhance the rate of carbothermal reduction due to their large specific surface areas and excellent activity, resulting in a rapid increase in the partial pressure of SiO and CO inside the char layer. In addition to that, the MWCNT network structure in the char has a directional effect on in situ SiC (silicon carbide) formation, which improves the ablation resistance of the nanocomposite. In this sense, higher amounts of SiC can contribute to some challenging applications in the automotive and aerospace industries. To tailor the thermomechanical properties of EPDM-based nanocomposites, Iqbal et al. [115] incorporated silane-treated MWCNTs (S-MWCNTs) into an EPDM matrix using five different mass concentrations: 0, 0.1, 0.3, 0.5, and 1.0%. Using TG/DTG analysis, the authors observed that the thermal stability increased as the amount of S-MWCNTs in the EPDM matrix increased. This is due to the high thermal endurance and nanoscale interaction of the nanotubes, which restricts the thermal motion of the molecular polymer chains in the heating environment [115]. In addition, two main mass loss events were observed for the nanocomposites in the temperature range of 200 to 520 °C. The first event in the 200–450 °C temperature range was associated with the evaporation of aromatic oil, wax, and other processing ingredients. The second event between 450 and 520 °C was attributed to EPDM pyrolysis [115].

### 5.2. DSC Analysis

Dynamical Scanning Calorimetry (DSC) is a type of thermal analysis in which the heat flow from the sample and the reference material is measured (energy absorbed or released) as a function of temperature during heating, cooling, or at constant temperatures [110]. The DSC curve shows the exothermic and endothermic effects, specific heat capacity, reaction and transition enthalpies, and other effects during temperature variations [110].

DSC measurements are widely used in the thermal analyses of materials to evaluate the transition temperature of exothermic and endothermic events and the specific heat capacity of materials. Jeon et al. [116] studied the effect of tungsten bronze nanorod and nanoparticle (TBNR or TBNP) loading on the *T_g_* of EPDM using DSC analysis, as shown in Figure 4. Second-order transitions in the DSC thermogram with *T_g_* values equal to −48.3 °C and −47.3 °C for neat EPDM and EPDM/TBNR (3 wt.%), respectively, were observed; for the 3 wt.% EPDM/TBNP, a *T_g_* of −46.5 °C was observed [116]. Only the second-order transition peak was observed in the DSC thermogram, indicating that the EPDM-based nanocomposite filled with TBNRs or TBNPs has an amorphous structure [116].

Iqbal et al. [115] investigated the influence of S-MWCNTs on the glass transition (*T_g_*), crystallization (*T_c_*), onset melting (*T_m_*_1_), and peak melting (*T_m_*_2_) temperatures of an EPDM-based nanocomposite. They observed that the *T_g_* value was reduced with the incorporation of S-MWCNTs in the EPDM matrix; the *T_g_* value of neat EPDM was −10.35 ± 0.10 °C and that of the EPDM-based nanocomposite with 1.0 wt.% S-MWCNT was −20.01 ± 0.10 °C [115]. By contrast, the opposite was observed for *T_c_*; the *T_c_* values were 4.55 ± 0.10 °C and 11.82 ± 0.10 °C for neat EPDM and the EPDM-based nanocomposite containing 1.0 wt.% S-MWCNT, respectively. These results were attributed to the formation of a nanofiller network in the EPDM matrix that restricts phase changes by interacting with the polymeric chains [115]. *T_m_*_1_ and *T_m_*_2_ were improved up to 58.28 °C and 11.21 °C, respectively, due to the incorporation of S-MWCNTs, which resisted polymer chain mobility by entrapping phonons within the nanofiller network. Additionally, incorporating S-MWCNTs resulted in higher thermal stability [115].

DSC analyses were used to investigate the *T_g_* behavior of EPDM with incorporated montmorillonite (MMT) nanoclay [112]. The results obtained by Rana et al. [112] showed that an increase in the amount of nanoclay increased the *T_g_* value of the EPDM-based nanocomposite, i.e., the *T_g_* values were −62.7 °C, −58 °C, −47.3 °C, and −33 °C for neat EPDM, and 2 wt.%, 4 wt.%, and 6 wt.% for nanoclay, respectively. Zhang et al. [113] also obtained similar results for an EPDM/nanoclay nanocomposite with a crosslinking interfacial design. The study was conducted from –80 °C to 80 °C, and the authors observed an increase in the *T_g_* value of the EPDM-based nanocomposite as the amount of nanoparticles amount increased [113]. The *T_g_* value of neat EPDM was equal to −53.5 °C, which increased to approximately −49 °C with the incorporation of 10 phr (per hundred rubber) in the EPDM matrix [113].

### 5.3. DMTA Analysis

The DMTA technique is a type of thermal analysis that involves the application of periodic tension and evaluating the sampling effort with temperature variation, i.e., the study of stress can be performed as a function of temperature or frequency of minor variable mechanical stress [110]. From DMTA analysis, it is possible to calculate the complex modulus (*E*′ and *E*″), tan *δ*, and the viscoelastic parameters of materials. During cyclic strain over a temperature range, the complex storage modulus *E** can be written as shown in Equation (1):(1)$$E*=E^{\prime }+E^{\prime \prime }$$

*E*′ represents the storage modulus, which can also be defined as the quantity of energy stored and retained during the load application cycle; *E*″ is classified as the loss modulus or energy dissipated during the cycle. tan *δ* can be calculated using the *E*″/*E*′ ratio, as shown in Equation (2):(2)$$\tan\delta =\frac{E^{\prime \prime }}{E^{\prime }}$$

Most of the materials developed by the industrial and technological sectors are subjected to dynamic deformations/strain and temperature variations during their life cycles. In this sense, it is essential to investigate the dynamic mechanical properties during the temperature variation of polymer nanocomposite and analyze the modes of cyclic deformation and relaxation processes similar to those of temperature.

DMTA measurements were carried out on neat EPDM and EPDM-based nanocomposite reinforced with silica (SiO_2_) nanoparticles (Figure 5) to evaluate *E*′, *E*″, and tan *δ* at temperatures ranging from −100 °C to 100 °C [117]. The behavior of *E*′ and *E*″ as a function of temperature shows two well-defined regions, i.e., above (rubbery region) and below (glassy region) the *T_g_.* For *E*′ below the *T_g_* (glassy region), an increase of up to 10 wt.% of SiO_2_ was observed in the EPDM matrix, which indicates an increase in stiffness and restricted polymeric chain movement; on the other hand, a decrease in the quantity of SiO_2_ was observed [117]. In the *T_g_* region, the *E*′ value tends to decrease for all EPDM-based nanocomposite and neat EPDM, indicating decreased energy stored at temperatures superior to *T_g_* [117]. Above *T_g_* in the rubbery region, the *E*′ value was higher for samples with 30 and 20 wt.% SiO_2_; according to Mokhothu et al. [117], this behavior may be related to the degree of crosslinking in terms of the EPDM matrix and the content of the rigid dispersed phase. *T_g_* was determined using tan δ as a function of temperature at the glass transition maximum peak; the values were −41.5 °C and −42.3 °C for neat EPDM and the EPDM/SiO_2_ nanocomposite with 30 wt.% SiO_2_, respectively [117]. A slight increase in the *T_g_* value was observed for samples containing SiO_2_ due to the reduced mobility of the EPDM chain in the nanocomposites because the EPDM chains strongly interact with the SiO_2_ nanoparticles, decreasing polymer chain mobility [117]. In another work, the authors observed similar behavior when studying in situ generated silica nanoparticles on EPDM, in which they evaluated the morphology, thermal, thermomechanical, and mechanical properties of EPDM-based nanocomposites [118].

Basu et al. [119] reported the effect of fire-safe and environmentally friendly nanocomposites based on layered double hydroxides (LDHs) and an EPDM matrix, which can be applied to many products, e.g., appliance hoses, radiator hoses in cars, washers, and insulation for water pipes and outdoor appliances. Through DMTA analysis, the authors observed good EPDM–nanofiller interactions due to a positive shift in the *T_g_* (peak height of the tan *δ* vs. temperature) of the EPDM-based nanocomposite when the loading of LDH was increased from 4 to 100 phr, i.e., the *T_g_* value was equal to −42.4 °C and −36.8 °C for EPDM—4 LDH (4 phr) and EPDM—100 LDH (100 phr) nanocomposites, respectively [119]. Additionally, this work showed a decrease in the peak height of tan *δ* with increasing LDH quantity. This behavior indicates the reinforcing characteristics of LDH in the EPDM-based nanocomposite [119]. The *E*′ value as a function of temperature also presents an improvement with an increasing amount of LDH in the vitreous and rubbery regions; however, above *T_g_*, the *E*′ value decreased as the temperature increased for all samples [119]. This behavior of *E*′ can be attributed to the excess amount of LDH, which provides a higher aspect ratio and a greater exposed surface area.

## 6. Morphological Properties

The morphological characterization of polymers is essential as studies on polymer structure and morphology provide insight into the macroscopic, morphological, and conformation properties that depend on the composites or nanocomposites used as fillers in their composition [120]. There are several microscopy techniques, such as scanning electron microscopy (SEM), transmission electron microscopy (TEM), and atomic force microscopy (AFM), that can be used to characterize polymers, such as EPDM. Morphological analysis allows for the determination of changes in the structures and properties of polymers caused by the addition of nanofillers. These changes include the dispersion of charges [2], micro-deformations [121,122], topographical changes [33,87], and modifications in the self-organization of a polymeric structure [114,123]. For example, Table 1 shows some observations obtained via the morphological analysis of EPDM matrices modified with various nanofillers or nanoparticles.

SEM is an analytical tool that enables the qualitative characterization of polymer surfaces and is used to identify characteristics such as the topography [35,87], morphology [35,77,124,125], and composition of materials. SEM studies are essential because the mechanical and thermal properties of a polymer depend on the type of filler dispersion used in the polymeric matrix [2,84].

Mohamed Bak et al. [126] studied the effect of nanoclay loading on blends of EPDM/silica–styrene–butadiene–rubber (S-SBR) nanocomposites. The fractured surface of the nanoclay-filled EPDM/S-SBR rubber blends was investigated using SEM images. The SEM observations showed that the fractured surfaces containing several tear lines, matrix crack lines, crack initiation, and crack propagation, as well as high nanoclay loading (10.0 phr), are the reasons for the formation of clusters and its contribution to the poor dispersion of the nanofiller in the EPDM matrix. Azizi et al. [123] studied EPDM and EPDM/silicone rubber composites with additives of modified fumed silica (MFS), titanium dioxide (TiO_2_), and graphene. The SEM micrographs showed that the addition of graphene as an additive in the formulation of the composites facilitated the formation of MFS and TiO_2_ particles that were homogeneously dispersed in the EPDM matrix.

**Table 1 polymers-16-01720-t001:** Some observations were obtained by morphological analyses with different nanocomposite charges added to the EPDM matrices.

Matrices	Nanofiller	Observation	Ref.
EPDM/Silica-SBR	Nanoclay	High loading of nanoclay (10 phr)—formation of clusters with poor dispersion	[2]
EPDM	Organoclay	3 wt.% of fillers to the matrix cause a substantial wettability of elastomer	[35]
EPDM	MWCNT	The densities of samples increased as a function of MWCNT content	[28]
PP/EPDM	MWCNT	Morphological changes induced by the dynamic vulcanization process	[127]
EPDM/MFS	Graphene	The addition of graphene improved the homogeneous dispersion with a decrease in agglomerates.	[123]
EPDM/PP	Nanoclay and nanosilica	AFM and SEM show two phases and the uniform distribution of the nanofillers	[125]
SAN/EPDM	Organically modified montmorillonite	Micrographs indicate two phases (SAN + EPDM) with different domain sizes and shapes.	[124]
PP/EPDM	MWCNT	SEM and TEM show MWCNT aggregates in the matrix	[128]

Bizhani et al. [28] developed nanocomposites of EPDM rubber and MWCNTs from hot compression molding using a chemical blowing agent as a foaming agent. Figure 6 shows the SEM micrographs of the EPDM/MWCNTs containing different amounts (phr) of MWCNTs. The presence of MWCNTs in EPDM caused an increase in density owing to a decrease in the cell diameters from 227 ± 58 μm to 59 ± 30 μm [28]. According to the authors, the MWCNTs acted as nucleating sites during the foaming process, increasing the number of cells and decreasing the size of the cells.

TEM is an effective and adequate analytical method for conducting the morphological characterization of several types of polymer samples on a nanoscale, such as EPDM matrix polymers with nanometric fillers [45,128,129].

Jha and coauthors [45] studied oil-extended EPDM filled with nanographite. The morphological characterization conducted via SEM and TEM showed good dispersion when nanographite concentrations of up to 4% were utilized in the EPDM matrix; TEM was performed on the EPDM/nanographite samples with filler concentrations of 2%, 4%, and 6%. The nanofiller with up to 4% concentration was uniformly distributed in the EPDM matrix, and the EPDM elastomer was situated between the nanofiller.

Bhattacharya et al. [125] developed thermoplastic vulcanizates (TPVs) and TPV nanocomposites (TPVNs) based on EPDM and PP. The TPVNs were prepared with a fixed EPDM:PP ratio; the nanofiller content varied with the use of different concentrations. The AFM and SEM images showed that the nanofillers (nanosilica and nanoclay) were well distributed in the TPV matrix. Specifically, the AFM analysis made it possible to view the surface topography and surface heterogeneity of the different polymeric systems. The authors attributed the brighter regions to softer materials (lower nanofiller concentration) and the darker areas to harder materials (higher nanofiller concentration). The AFM images showed that the material exhibited a biphasic morphology, which corresponded with the SEM results of the same sample. A uniform distribution of nanofillers was observed in the samples filled with 1.0 phr nanofiller [125].

Vishvanathperumal and Anand [80] examined the microstructural characteristics of EPDM/SBR hybrid composites. Within these composites, reinforcement was achieved by incorporating nanoclay and nanosilica particles. This addition facilitated enhanced properties attributed to the improved interaction between the fillers and the rubber matrix. The concentration of nanosilica and nanoclay emerged as a crucial factor determining the composite’s properties. Additionally, composites featuring agglomerated nanosilica exhibited superior wear resistance compared to other formulations within the study [80].

According to Rostami-Tapeh-Esmaeil et al. [81], rubber foams that undergo a more extensive procuring process exhibit thicker cell walls. As the procuring level increases, there is a noticeable trend toward a more confined range in cell sizes within the foam structure. Specifically, at a pre-curing level of 30, there is a greater quantity of cells than other pre-curing levels examined.

In Moustafa et al. [35], SEM images depicted proofed fabrics filled with different loads of EOC and OC. The effective dispersion of the organoclay nanofiller played a crucial role in preventing radiation-induced degradation of the polymer matrix. Additionally, incorporating organoclay notably enhanced the resistance of filled PEPA blends against photo-oxidation.

In an isotropic Magnetorheological Elastomer (MRE) presented by Kang et al. [130], carbonyl iron particles were evenly and haphazardly distributed within the elastomeric matrix. Conversely, in an anisotropic MRE, these carbonyl iron particles were aligned within the elastomeric medium in a specific direction or pattern.

The SEM images indicate a consistent and even distribution of MWCNTs within the EPDM composites in the work of Gou et al. [49]. Additionally, a TEM image validated this uniform dispersion of MWCNTs within the composite structure. Moreover, the SEM images illustrated the formation of a char layer, specifically in composite 3, which was within the high-speed area.

Chen et al. [90] proposed that EPDM biocomposites filled with silane-modified wood fibers (Si-WFs) should be manufactured. This study investigated the impacts of fiber loading, silane modification, and crosslinking. Silane modification notably enhanced water absorption resistance and thermal stability. Applying silane modification led to an enhancement in Si-WF dispersion within the elastomer matrix. This improved dispersion of Si-WFs within the matrix was confirmed through fractography analysis [90].

Observing the morphology in the work of Chang et al. [83] supports creating an EPDM network that percolates rheologically. Solid-state fibrils inhibit molecular mobility and constrain relaxation when subjected to shear flow. Additionally, the microstructure of blends was examined following partial etching. Burgaz and Goksuzoglu [85] noted a uniform dispersion of individual CIP particles alongside their clusters. Conversely, BIP particles were observed creating larger clusters. The EPDM/CB/CIP MREs displayed noteworthy enhancements in terms of their properties compared to EPDM/CB/BIP MREs.

Abdelsalam et al. [91] showed that fillers such as CB particles play a pivotal role in augmenting the physical attributes of rubber-based products. The dispersion of CB and its interaction within the rubber matrix play significant roles in determining the resulting mechanical properties. Increased CB loading correlates with elevated torque levels and enhancements in tensile strength, tear strength, and crosslink density. SEM images illustrated an adequate dispersion of CB particles within the rubber matrix. Furthermore, the surface achieved homogeneity upon reaching a CB loading of 45 phr [91].

## 7. Thermal Conductivity

The thermal conductivities of polymers, such as elastomers, are relatively low, with values of 0.1–0.2 W/(m K). Therefore, loading nanofillers, such as metal particles, graphene, and CNT, into polymeric matrices can improve thermal conductivities [131]. In a study reported by Bizhani et al. [43], a highly elastic foam with enhanced electromagnetic wave absorption made of EPDM filled with a barium titanate (BT)/MWCNT hybrid was produced. The EPDM-based nanocomposite foam demonstrated enhanced thermal conductivity with an increase in the quantity of MWCNTs, i.e., the hybrid EPDM foam exhibited a higher value [0.217 W/(m K)] compared to the sample composed of only 20 phr BT [0.077 W/(m K)], as illustrated in Figure 7 [43]. This behavior is attributed to an improvement in the thermal conduction through the solid and gas phases of the foam [43]. According to the authors, the EPDM-based nanocomposite can be a valuable asset in electromagnetic interference (EMI), shielding applications owing to its improved EM wave absorption due to heat dissipation and minimizing the impact of temperature on the electronic components [28,43].

Lu et al. [29] developed a flexible GnPs/EPDM composite with excellent thermal conductivity and EMI shielding properties. The amount of GnPs incorporated into the EPDM matrix greatly affected its thermal conductivity, i.e., as the amount of GnPs in the EPDM-based nanocomposite increased, the thermal conductivity was enhanced. This behavior indicates that the GnPs assisted in improving the thermal conductivity inside the EPDM matrix; the thermal conductivity values were 0.23 W/(m K) and 0.79 W/(m K) for neat EPDM and the EPDM-based nanocomposite with 8 wt.% GnPs, respectively. The results are significant for potential applications in sensors, as heat dissipation affects the stability and sensitivity of sensors [44]. The percolation threshold was reached at 2.9 wt.% graphene platelets (GnPs), significantly increasing electrical conductivity. By increasing the concentration of GnPs to 7 wt.%, the thermal conductivity improved significantly, reaching 0.72 W/m K, nearly three times higher than at 2.9 wt.%. As a result of the improved dispersion of GnPs within the EPDM matrix, thermal conductivity is improved. In the sensor with 3 wt.% GnPs loading, the gauge factor was 129.33, indicating excellent sensitivity. With a 7 wt.% GnPs loading, the sensor exhibited enhanced sensitivity, with two linear change sensing stages, each with a different gauge factor, implying improved operating stability and sensitivity. Higher thermal conductivity resulting from the increased GnP content allows sensors to dissipate heat more efficiently, resulting in excellent stability and sensitivity.

## 8. Electrical Properties

As previously mentioned, dispersing nanofillers or nanoparticles in a polymeric matrix to produce nanocomposites offers several advantages regarding the morphology, thermal properties, and mechanical properties of materials. Additionally, the addition of nanofillers with excellent conductive properties can provide an additional benefit, improving the electrical properties of nanocomposites and, thereby, expanding their potential applications [132,133,134]. The combination of conductive nanofillers with polymeric matrixes provides unique and highly desirable electrical properties, making conductive nanocomposites a fascinating class of materials [135]. These conductive nanocomposites have several applications, ranging from flexible electronics to advanced sensors and energy storage devices, such as supercapacitors [135,136,137].

To develop new materials for desired applications, it is essential to understand the conduction processes and phenomena that govern electrical conductivity in conductive polymer nanocomposites [9,138,139]. The percolation theory is one of the widely used theories for understanding conduction processes and electrical conductivity in polymer nanocomposites [140,141]. According to this theory, the electrical conductivity of a composite material increases abruptly as the volume fraction of the conducting nanofillers reaches a certain critical fraction or percolation threshold [9,139,140]. Percolation thresholds may be defined as the critical concentration (mass or volume) of conductive nanofillers dispersed in an insulating matrix [9,139,140]. When the concentration of conductive nanofiller reaches the percolation threshold in the insulating matrix, the first conductive path is formed by which the charge carriers move under the action of the external electric field [9,139,140]. In other words, when concentrations of conductive nanofillers reach the percolation threshold, an insulator–conductor transition occurs in the nanocomposite, resulting in an abrupt increase in its electrical conductivity [9,139,140]. When the concentration of conducting nanofillers in nanocomposite is below the percolation threshold, they do not form conductive paths far away from each other, resulting in low electrical conductivity [9,139,140]. Above the percolation threshold, however, a continuous conductive path is formed that allows electrons to flow through the composite material [9,139,140].

In this sense, it is possible to understand how the dispersion of nanofillers influences the electrical conductivity of conductive nanocomposites by using the percolation theory [142,143]. For nanofillers to form effective conductive paths, homogeneous dispersion and good interconnectivity are essential [136,142,143]. Furthermore, the size, shape, and concentration of nanofillers are also crucial factors in determining the percolation threshold and the electrical conductivity of the nanocomposite [139,144,145]. Hence, the percolation threshold is defined as the critical volumetric fraction of conductive nanofillers in the nanocomposite, above which a continuous conductive path is formed [140,141,146]. Because of the considerable distance between the conducting nanofillers below the percolation threshold, no conductive paths are formed, resulting in an electrical conductivity like that of the insulating polymer matrix [9,138,139]. In contrast, above the percolation threshold, electrical conductivity increases dramatically due to the interconnection of the conductive paths, allowing electrons to flow efficiently, bringing the electrical conductivity of the nanocomposites closer to the conductive nanofillers [9,138,139].

Notably, dispersing conductive nanofillers in EPDM matrices has benefits. EPDM elastomers are widely used in applications requiring flexibility, chemical resistance, and durability [147]. Unfortunately, the low electrical conductivity of these materials often limits their use in electronic devices and in applications where good electrical conductivity is essential. It is possible to significantly improve the electrical properties of EPDM by incorporating conductive nanofillers, such as CB, CNT, graphene, metallic nanoparticles, etc. [145,148,149,150]. As a result of the uniform dispersion of these nanofillers, a three-dimensional conductive network is created within the material, allowing the charge carrier to transfer efficiently [148,150,151]. Increasing the volume fraction of the conductive nanofillers increases the electrical conductivity of the EPDM/nanofiller nanocomposite because a continuous conductive path is formed, as predicted by percolation theory [148,150,151]. In this way, EPDM can be modified and improved in terms of its electrical properties, making it suitable for a wide range of electronic and electrical applications, such as antistatic coatings and films, sensors, conducting adhesives, electromagnetic interference shielding materials, etc. [135,136,137,147].

The thermal and electrical conductivity of EPDM-based nanocomposites with GnPs and CB has been extensively studied by Koca et al. [148]. The electrical conductivity of EPDM/GnPs composites is slightly affected by the specific surface area or lateral size of GnPs up to 20 phr [148]. The electrical conductivity of EPDM-based nanocomposites was higher when hybrid fillers, 50 phr CBs, and 7 phr GnPs with higher lateral sizes were used. EPDM-based nanocomposites with a single CB have electrical conductivity increases of 4.4 × 10^15^ S/cm to 4.8 × 10^−6^ S/cm from 20 phr to 50 phr, according to the authors [148]. Electrical conductivity is enhanced by the increasing number of contacts between CB particles, which leads to more conductive pathways and improves conductivity. Moreover, the filler–filler interaction pathway facilitates electron transfer through the matrix, and hopping and tunneling are two mechanisms for electron conduction in polymer composites [148].

Zhang et al. [150] used CNT and hexagonal boron nitride nanosheets to improve the electrical properties of EPDM-based nanocomposites for cable accessory applications. Based on the results, it can be concluded that the non-linear conductivity of CNT/h-BN/EPDM composites becomes more prominent with increasing CNT content, followed by a reduction in threshold field strength and an increase in the non-linear coefficient [150]. In order to study the electrical properties of donor ZnO nanoparticles/EPDM composites, Chi et al. [149] developed EPDM-based nanocomposites that exhibit outstanding non-linear electrical conductivity. The results indicate that the non-linear conductivity becomes much more distinct with the increase in ZnO nanoparticles, along with an increase in the non-linear coefficient of conductance and a decrease in breakdown field strength [149]. Consequently, ZnO/EPDM nanocomposites demonstrate promise as an effective means of protecting the safe operation of power transmission systems [149].

## 9. EPDM-Based Nanocomposites: Advantages, Disadvantages, and Perspectives

EPDM rubber is a type of synthetic rubber that is used in a variety of applications. EPDM rubber is classified as a class M rubber by ASTM standard D-1418 [152], which includes elastomers of the saturated chain polyethylene type. In 2022, the market for EPDM is estimated to be worth USD 4.45 billion, and it is expected to grow at a compound annual growth rate (CAGR) of 5.7% between 2023 and 2030. The primary driver of the growth of the automotive industry is increasing investments in emerging economies such as China and India. There has been considerable expansion of application sectors, including automotive, building, and construction industries. In recent years, construction and automotive industries in the United States have experienced rapid growth, which has increased regional demand for products [153].

Because of their mechanical and thermal properties, EPDM rubber nanocomposites are highly promising for a variety of industrial applications. When nanoparticles are incorporated into nanocomposites, they significantly increase tensile strength, elastic modulus, and wear resistance, outperforming other nanocomposites under certain conditions [154,155]. Additionally, these nanoparticles enhance the thermal stability and oxidative degradation resistance of the materials, making them suitable for use in adverse environments [154,155].

Compared to SBR-based nanocomposites, EPDM-based nanocomposites exhibit superior heat and oxidation resistance, making them more suitable for extreme conditions [156,157]. EPDM is highly resistant to heat and oxidation, which makes it ideal for applications that require durability under severe conditions [158,159]. When compared to silicone rubber, EPDM offers a unique combination of flexibility and mechanical strength. Despite the fact that it cannot reach the extreme temperatures of silicone, EPDM is more cost-effective.

The versatility and weather resistance of EPDM nanocomposites make them stand out among synthetic elastomers. However, achieving a homogeneous dispersion of nanoparticles can be challenging, negatively impacting the final properties and causing production costs to increase, which limits the application of nanoparticles at a large scale. Rubber-based nanocomposites, such as EPDM-based nanocomposites, offer a balance between price and performance, but the synthesis and incorporation of nanoparticles can increase the production cost. Nanocomposites can have adverse environmental effects due to the presence of nanoparticles, which are difficult to recycle or degrade [160,161].

Depending on the specific application requirements, EPDM rubber nanocomposites are preferred over other materials. In general, these materials are selected for applications that require high flexibility, wear resistance, and thermal stability, such as automotive components and building materials. In contrast, epoxy- and polyamide-based nanocomposites may be more suitable for applications requiring high mechanical strength and rigidity [147,162]. Table 2 summarizes the advantages and disadvantages of specific elastomeric materials.

**Table 2 polymers-16-01720-t002:** Advantages and disadvantages of elastomeric materials.

Matrix	Advantage	Disadvantage	Ref.
EPDM	Increased tensile strength, elastic modulus, wear resistance; superior resistance to heat and oxidation; high mechanical strength and flexibility; cost-effectiveness	Nanoparticle dispersion challenges, increased production costs, environmental concerns	[126,163,164,165,166,167]
SBR	Cost-effective; good abrasion resistance; excellent adhesion to fabrics	Heat and ozone resistance are low; oil and solvent resistance is low	[168,169]
Natural Rubber	Excellent tensile strength; high elasticity; high resilience	The product has a low resistance to heat, ozone, oils, and solvents	[161,170,171]
Silicone Rubber	Heat and cold resistance; good electrical properties; chemically inert	High cost; low tensile strength	[172,173,174]
Synthetic Rubbers (General)	Adjustable properties: more excellent resistance to oils, solvents, and weather than natural rubber	It may be less elastic and resilient than natural rubber; some types may be more harmful to the environment.	[154,155]
Epoxy- or Polyamide-based Nanocomposites	Mechanical strength and rigidity are high	Comparatively less flexible than elastomeric materials	[147,162]

Figure 8a illustrates the effects that different types of nanofillers can have on EPDM-based nanocomposites and how they are intended to affect their properties. Figure 8b illustrates a simple representation of the microstructure of different types of nanofillers, including 0D, 1D, and 2D fillers.

Furthermore, it is essential to be aware of the main trends in terms of improving the properties, processes, and load systems of this elastomer. In recent years, there has been a growing interest in improving the energy absorption properties of natural-fiber-reinforced polymer composites, especially in structural applications. This study focuses on developing PP/EPDM composites based on coconut fibers (10–40 phr), which have been plasma pretreated and coated with highly hydrophobic fluoroalkyl functional siloxanes. A natural fiber elastomer is an example of the trend toward proposing elastomers with high sustainability for technical applications [175]. In addition, a foam composite was developed with bamboo fiber, polypropylene, and EPDM, with an elastomer content of 10%. This material demonstrated the best foaming effects, and its impact toughness increased by 34.42% compared with pure polypropylene. This study expands the possibilities of using bamboo fiber-reinforced polypropylene microcellular foam materials [176].

This exercise illustrates how a mixture can be produced with the required properties and cost-effectively, as opposed to producing samples, which does not energy-consuming equipment without which development would not be possible and that has an ever-increasing value, without which the sample could not be produced and the results studied. According to the results of this study, the equivalent properties of the composite studied were 23.0% in terms of rebound resilience, 57.8% in terms of elongation at break, 7.8 MPa in terms of tensile strength, and 66.8% in terms of Shore A hardness. Particulate load optimization techniques and statistical design of experiments have been demonstrated to be effective methods for generating compositional variables when combined. As a result, there is potential for the development of new analytical processes for elastomers in the future, such as EPDM, as well as other applications that utilize machine learning tools to validate and predict the development of new elastomer blends, including [177,178,179]. Furthermore, in the development of thermoplastic elastomers (TPVs), the mixtures EPDM/PP, the higher crystallinity of the PP in the TPV blend, the higher Mooney viscosity, as well as the better compatibility between EPDM and PP, result in higher tensile strength and elongation at break, which are enhanced by the crystals present in EPDM [180]. Therefore, EPDM is an elastomer with a long way to go in terms of new applications relating to viscosity and rheology, as in this case for the application of TPV. Table 3 shows the correlation between the EPDM matrix or EPDM-polymer blend and how different nanofillers affect the microstructures, as well as their effect on the properties of EPDM-based nanocomposites.

**Table 3 polymers-16-01720-t003:** Effects of nanofiller loading on EPDM-based nanocomposite microstructure and properties.

Matrix	Nanofiller	Morphology	Properties	Ref.
EPDM-SBR	Nanoclay	A high loading of nanoclay (10 phr) results in the formation of clusters with poor dispersion	As the concentration of nanoclay (4–10 phr) is gradually increased in the EPDM/S-SBR rubber nanocomposites, the 100% modulus of elasticity increases. In the same manner, when nanoclay loading is up to 8 phr, the compression set increases, but as the load increases, the compression set decreases due to the afferent crosslinking density and mobility of the long rubber chains. From abrasion resistance to hardness, nanocomposites showed a gradual increase in both properties.	[2]
EPDM	Organoclay	The presence of 3 wt.% of fillers in the matrix makes the elastomer more wettable.	Thermal degradation of pure EPDM was observed using TGA. The maximum degradation temperature (*T_max_*) at 10% weight loss of the material was 478 °C, indicating high thermal stability when compared to other studies. The onset temperature of degradation (*T*_10%_) at 10% weight loss of the material was 401 °C. Antimicrobial tests were conducted on tissue samples coated with 5% by weight of EOC or OC and 20% by weight of a long oil alkyd resin based on soybean oil. It was conducted against three different types of microbes: Staphylococcus aureus (G-ve bacteria), Pseudomonas aeruginosa (G-ve bacteria), and Candida albicans (fungi). LAR resin (fatty acid) with expanded organoclay is likely to serve as an essential component of microbial fatty acids that are a promising target for the development of antimicrobial-resistant tissues in the presence of 20 wt.% LAR resin (fatty acid).	[35]
EPDM	MWCNT	Sample density increased with MWCNT content	The development of EPDM/MWCNT foams has achieved superior EMI-specific SE and deformability. The use of EPDM should also ensure properties such as chemical, moisture, and ozone resistance. The developed foam samples exhibit high thermal and electrical conductivities of up to 2.7 × 10^−4^ S/cm and EMI shielding efficiencies of up to 45 dB, which do not degrade significantly after repeated bending. The EPDM matrix exhibits these properties because of the formation of a three-dimensional interconnected network.	[28]
PP/EPDM	MWCNT	Dynamic vulcanization causes morphological changes	Despite a high loading level, the samples have formed a dense, stable conductive network, which is difficult to destroy, resulting in excellent resistance stability. The cyclic loading test revealed no fractures and electrical shots, which indicates strong potential for practical applications. Approximately 0.1% of incident electromagnetic waves could penetrate the material without being reflected or absorbed, primarily as a result of the high density of the MWCNT network and the greater electrical conductivity of the samples compared to TPE without filler, indicating that the TPV elastomer composites may be helpful to as materials for stretchable conductors with electromagnetic radiation shielding capabilities.	[127]
EPDM/MFS	Graphene	With the addition of graphene, homogeneous dispersion was improved, and agglomerates were reduced.	EPDM/S/TiO_2_ samples showed a 30% increase in dielectric strength over vulcanized EPDM rubber samples. The addition of MFS, TiO_2,_ and graphene to EPDM and EPDM/S rubbers resulted in a higher thermal conductivity for all composites. The EPDM composites containing TiO_2_ had the highest thermal conductivity values. The thermal conductivity of the composites did not appear to be affected by the addition of graphene.	[123]
ultra-high molecular weight EPDM (UHMW-EPDM) and PP	Nanoclay and nanosilica	SEM shows the presence of two phases and a uniform distribution of nanofillers	Compared with the other blends, the tensile strength and modulus were 250% higher with the 7 phr nanofiller loading. Nanosilica-added TPVs are superior to nanoclay-added TPVs in terms of mechanical properties, crosslink density, and morphology.	[125]
SAN/EPDM	Organically modified montmorillonite	Micrographs indicate two phases (SAN-EPDM) with different domain sizes and shapes.	Incorporating nanoclay and EPDM improved the thermal stability of SAN. Due to the establishment provided by EPDM rubber (known for its high thermal stability) and the improved interaction between SAN and nanoclay, nanocomposite blends demonstrated higher thermal stability than pure SAN. Because the EPDM soft phase has a low modulus, the Young’s modulus of the binary blends was decreased. As long as there is good phase interaction, adding a low-modulus EPDM dispersed phase to a high-modulus matrix (pure SAN) will tend to reduce its modulus.	[124]
PP/EPDM	MWCNT	TEM and SEM images show aggregates of MWCNTs	There was a significant difference between the unvulcanized mixtures and the vulcanized TPVs, which had superior tensile strength and deformation resistance. Dynamic vulcanization played a substantial role in determining the fracture toughness of vulcanized samples, although different curing mechanisms contributed to fracture toughness in various ways.	[128]

The development of porous systems has been one of the prospects for the application of EPDM-based nanocomposites, which have been applied to a variety of industrial contexts, ranging from household products, acoustic systems, thermal protection, and aerospace applications. Ohadi, Asghar, and Hosseinpour [181] have shown the application of MWCNT on EPDM foams to absorb low-frequency waves. Samples with low MWCNT loading showed higher damping factors, and smaller pore sizes exhibited high viscous motions with superior sound absorption characteristics. Thus, it was possible to know that the nanocomposite foam sample with only 0.1 phr (parts per hundred of rubber) of MWCNT offered the best sound absorption coefficient with a sharp peak (0.98) in the frequency range of 500–1000 Hz. This occurs due to the high porosity, the airflow resistance of the thin, and the partially crosslinked cell structure. With additional MWCNT loading, the cell size was drastically reduced, resulting in a decrease in porosity and an excessive increase in airflow resistance. It can be concluded that the MWCNTs improve the sound absorption and acoustic insulation performance of EPDM foam so that with the application of nanoparticles, a lightweight acoustic absorber with high resistance to adverse environmental conditions and excellent low-frequency sound absorption can be obtained.

To improve foams, biobased materials can reduce the cost of commercial fillers, change the degradation lifetime, and add environmentally safe credits that offset the price of the new material. EPDM rubber-based biocomposites filled with silane-modified wood fiber (SiWF) were foamed using the chemical foaming method by Chen, Gupta, and Mekonnen [182]. Using SiWF and dicumyl peroxide (DCP), the developed EPDM biocomposite foams showed much lower water absorption and improved thermomechanical properties. The tensile strength and modulus of the EPDM biocomposite foams exhibited up to 90% and 600% enhancements with SiWFs and in situ crosslinking [182]. Figure 9 illustrates the EPDM biocomposite in foam form, as shown by the authors [182]. There was a high degree of foaming in the unfilled EPDM. In contrast, biocomposites are less foamed due to the higher concentration of fillers, silane modification, or DCP crosslinking [182].

The developed biocomposite foams also displayed lower apparent thermal conductivity than the unfilled EPDM foam over the heating temperature range, showing possible applications as insulation materials. Developing lightweight insulating materials has also proven to be an effective means of improving the performance of solid rocket motors (SRMs) [183]. Through rubber foaming technology, Wang et al. [183] introduced a porous structure into EPDM insulating materials to prepare lightweight insulating materials [183]. When studying morphology formation in the various samples, it was found that with an increase in pre-curing temperature, the cell size decreased significantly, and the original cell was more uniform, which improved the mechanical properties [183]. With an increase in blowing agent content, the cell density gradually increased without a significant change in cell diameter. During this period, the mechanical properties of the insulation materials gradually decreased [183]. EPDM rubber foams have the highest rate of carbonization resistance when the pre-curing temperature is 120 °C and the blowing agent content is 6 phr [183]. As compared to the base formulation, their carbonization ablation rate is only 16% higher. As a result, CNTs and carbon fibers (CFs) were added to reinforce the EPDM rubber foams [183]. Compared to the foam formulation, the reinforced formulation exhibited a 36.4% increase in tensile strength, which is 19.2% lower than the base formulation [183]. In comparison to the foam formulation, the carbonization rate of the reinforced formulation decreases by 35.5%, which is only 19.2% lower than the base formulation [183].

Zhao et al. [184] showed that polylactic acid (PLA) is a promising biomass and biodegradable polymer, but it is brittle and has a poor foaming ability. Herein, a composite with UV-crosslinking-assisted in situ fibrillation of PLA reinforced with ethylene-propylene-diene terpolymer (EPDM) nanofibrils was developed. Due to the heterogeneous nucleation effect of UV-crosslinked EPDM nanofibrils, the crystallization rate of PLA was significantly increased. The crystalline morphology was refined, resulting in a refined crystalline morphology with fibrous structures, and the tensile and impact strength were significantly improved without sacrificing strength and stiffness. In addition, the expansion ratio and cell population density of the foams increased by 470% in the presence of UV-cured EPDM nanofibrils. Due to the very high expansion ratio and the unique micro/nanoscale structures of the cell walls, the 28-fold-expanded PLA/EPDM foams exhibited excellent thermal insulation performance, with a thermal conductivity of only 26.5 µm.

On the other hand, Sang et al. [185] studied the properties of a bio-EPDM where Keltan^®^ Eco 6950C was produced from biological raw materials. In contrast, the ethylene used in the process is derived from ethanol produced from sugarcane. Foams prepared by mixing chemical and encapsulated foaming agents (OBSH (4,4′oxybis-(benzene sulfonyl hydrazide) at different mixing ratios were compared the results of sulfur crosslinking systems and dicumyl peroxide systems, with an evaluation of the mechanical properties, thermal stability, and saltwater resistance of the bio-EPDM foam. The mechanical and elastic properties of the bio-EPDM foam decreased with increasing amounts of encapsulated blowing agents. At the same time, the thermal stability and saltwater resistance improved significantly with increasing amounts of encapsulated blowing agents. The mechanical properties obtained with the peroxide system were superior to those obtained with sulfur. Further research focusing on sustainability, and no less important, involved the reuse of rubber (RR) in the development of closed-cell elastomeric foams based on ethylene-propylene-diene rubber (EPDM). Rheometric results showed that the introduction of RR up to 20 phr increased the cure rate from 11.7 to 13.48%/min, reduced the cure time from 12.21 to 9.3 min, and increased the final torque from 6.51 to 7.24 N·m. RR increased the cell density from 12 to 78 cells/mm^3^ and reduced the average cell number size from 940 to 110 µm. Furthermore, the introduction of RR proved to be a viable alternative to produce high-performance EPDM foams with improved toughness and resilience and predicts opportunities for EPDM/RR foam composites to be used in the sealing and gasketing industry as an environmentally friendly substitute for virgin rubber [186].

Nowadays, with the approach of computational tools for the optimization of processes and materials, the study of predictions and projections towards the design of composite systems has been considered. Buchen et al. [187] performed a study involving the time-dependent modeling and experimental characterization of foamed EPDM rubber. In their research, they implemented a derived constitutive model in finite element software (ABAQUS) and calibrated it experimentally with multi-step relaxation tensile tests of foamed EPDM rubber [187]. It is possible to predict the time-dependent stress–strain relationship for EPDM foam rubber using the viscoelastic model as a first approximation. A homogenized hollow sphere constitutive model described the hyperelastic behavior, while rheological elements described time dependence. Analyses were conducted using the finite element program Abaqus. By using a material without pores, it was possible to obtain information regarding the material’s behavior and mechanical characteristics, as well as multi-step relaxation experiments. In each case, the parameters were identified using least squares fitting. An equilibrium simulation resulted in the overestimation of stress even after optimizing parameter identification based on different deformation states. In addition to the analytical hollow sphere model, a numerical hollow sphere model was employed to evaluate the impact of matrix compressibility and to compare the predictions of the two models. Consequently, the numerical solution converges to the analytical model for increasing Poisson’s ratio, which opens the door for designing porous systems applicable to industrial applications.

EPDM-based nanocomposites can also be applied to the treatment of wastewater from the petroleum industry by using cellulose nanofibers to obtain nanocomposite membranes. According to Bhuyan et al. [188], a superhydrophilic and organic-solvent-resistant nanocomposite membrane was designed using waste bottles made from poly(ethylene terephthalate) (PET) and cellulosic paper. According to the study, 98 Lm^−2^h^−1^ permeability was measured and applied under 1.5 bar pressure [188]. Additionally, the membrane was able to remove more than 97% of the organic substances from a crude oil–water emulsion [188]. Moreover, EPDM-based nanocomposite foams or fibrous membranes can be used as membrane-based desalination technologies for the treatment of water, opening new avenues for water recovery [189].

## 10. Conclusions

EPDM is one of the most studied and used synthetic rubbers in the industry, and incorporating different types of nanofillers can expand its application in several areas. The dispersion of these nanofillers into EPDM plays an important role in improving thermal stability and conductivity. It is important to consider three aspects of filler dispersion. First, the surface chemistry of the nanofiller (presence or absence of active functional groups) may compromise the compatibility between the filler and the polymeric matrix. The second involves the interaction between the fillers and the polymer matrix, which can be chemical or physical and is dependent upon the formation of aggregates because of the interactions between the filler and the polymer matrix. Furthermore, the morphology of the particle should be considered as it should have a specific surface area of hundreds of square meters per gram, which can affect its dispersion and interaction with the polymeric matrix. Thirdly, it is important to select the correct polymeric matrix with which nanofillers and polymeric matrices must be compatible. Nanocomposites based on EPDM have several advantages over other rubber matrices, including their ability to accept large amounts of nanofillers that can greatly enhance their properties and extend their application range.

## Figures and Tables

**Figure 1 polymers-16-01720-f001:**
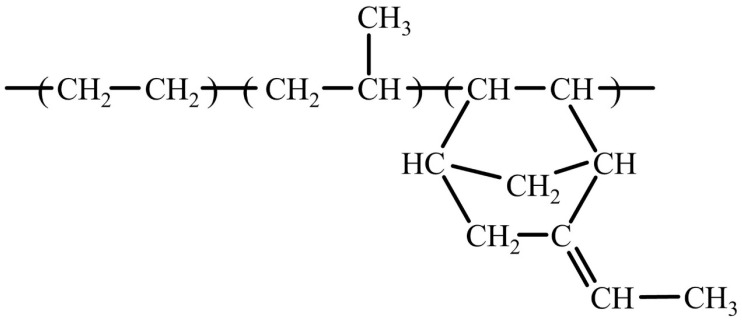
Representation of the chemical structure of EPDM.

**Figure 2 polymers-16-01720-f002:**
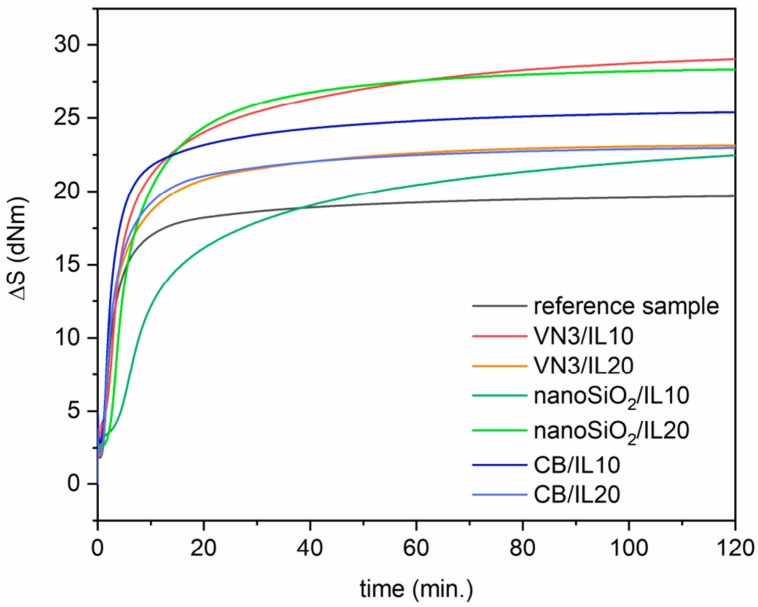
Rheometric curves of EPDM/SILPs compounds. (This figure is from ref. [79]).

**Figure 3 polymers-16-01720-f003:**
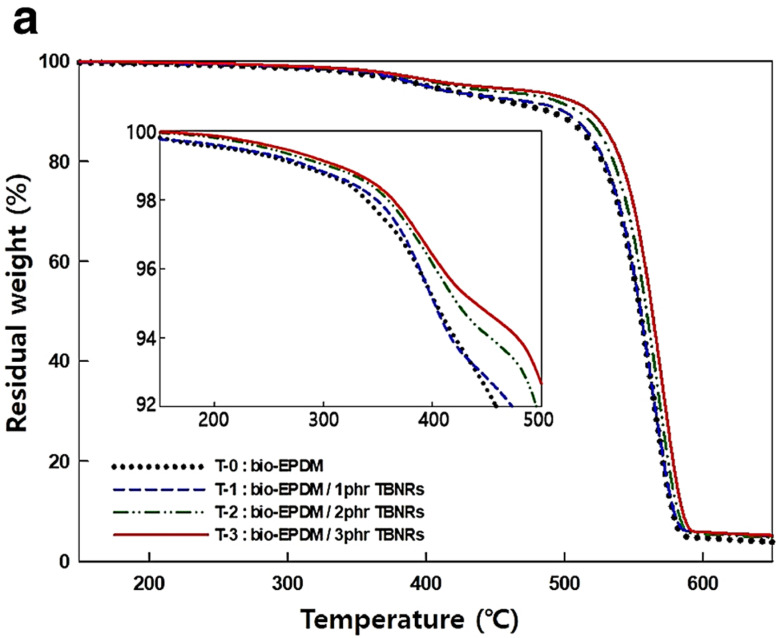
(**a**) TGA and (**b**) derivative TGA curves of bio-EPDM-based nanocomposite with various mass ratios of TBNRs. (This figure is from ref. [58] licensed under a Creative Commons Attribution 4.0 International License).

**Figure 4 polymers-16-01720-f004:**
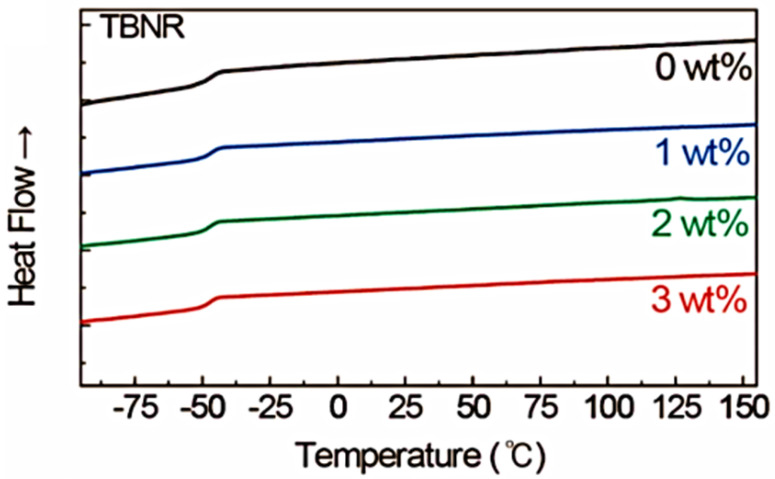
DSC curves of EPDM-based nanocomposite with various mass ratios of Na_0_._33_WO_3_ TBNR. (This figure is from ref. [116] licensed under a Creative Commons Attribution 4.0 International License).

**Figure 5 polymers-16-01720-f005:**
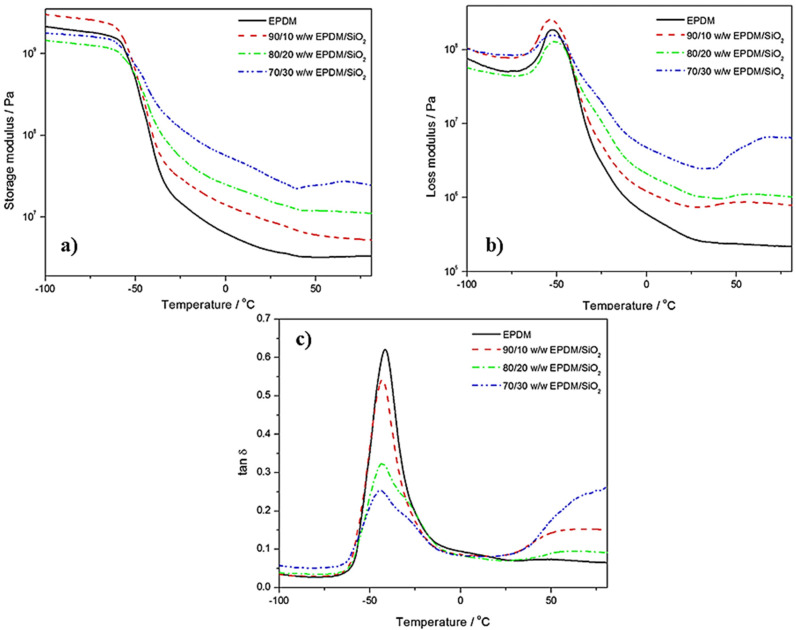
DMA curves of EPDM and EPDM/SiO_2_ composites: (**a**) storage modulus (*E*′), (**b**) loss modulus (*E*″), and (**c**) damping factor (tan *δ*). (This figure is from ref. [116] licensed under a Creative Commons Attribution 4.0 International License).

**Figure 6 polymers-16-01720-f006:**
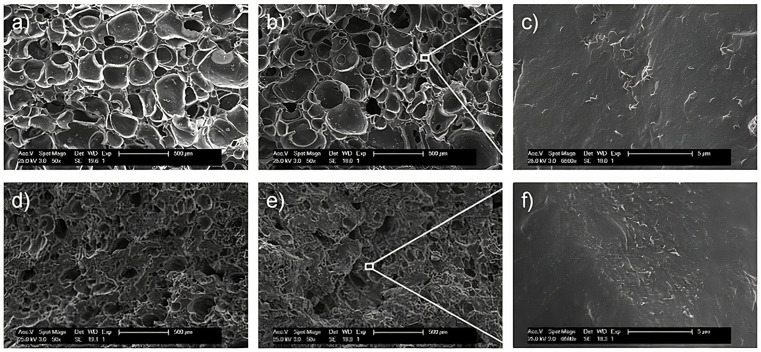
SEM micrographs of cryofractured EPDM/MWCNT nanocomposite foams containing (**a**) 0, (**b**) 2, (**d**) 6, and (**e**) 10 phr MWCNTs. (**c**,**f**) Micrographs of the MWCNTs within the morphology. (This figure is from ref. [28] licensed under a Creative Commons Attribution 4.0 International License).

**Figure 7 polymers-16-01720-f007:**
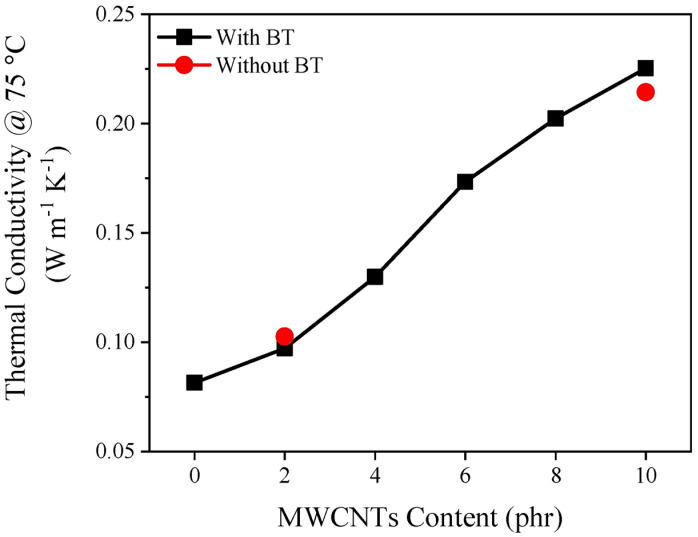
Thermal conductivity of cellular BT/MWCNTs and MWCNTs nanocomposites as a function of MWCNT loading at 25 °C. (This figure is from ref. [43] licensed under a Creative Commons Attribution 4.0 International License).

**Figure 8 polymers-16-01720-f008:**
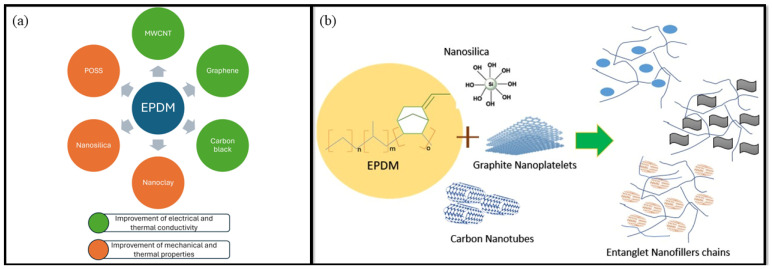
(**a**) Flowchart of EPDM-based nanocomposite with different nanofillers and their effect on its properties. (**b**) Microstructural representation of EPDM-based nanocomposite with 0D, 1D, and 2D nanofillers.

**Figure 9 polymers-16-01720-f009:**
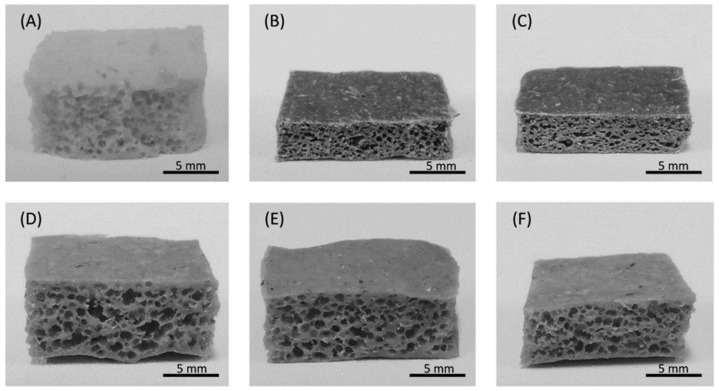
Images of the developed EPDM biocomposite foam. (**A**) Unfilled foam; (**B**) 40 WF; (**C**) 40 SiWF; (**D**) 10 WF; (**E**) 10 SiWF; (**F**) 10 SiWF-DCP. (This figure is from ref. [182] licensed under a Creative Commons Attribution 4.0 International License).

## Data Availability

No data were used to support this study.

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
