# Peer review of "A Review of EPDM (Ethylene Propylene Diene Monomer) Rubber-Based Nanocomposites: Properties and Progress"

_polymers, 2024, doi:10.3390/polym16121720_

Round 1

Reviewer 1 Report

Comments and Suggestions for Authors

Dear Authors,

The aim of this study is to provide readers with a general overview of the current state of Ethylene propylene diene monomer (EPDM) nanocomposites, the properties and applications were introduced. The topic is interesting and will be useful for interested parties. While the topic fits within the scope of the journal, there are some concerns that need to be addressed. Please consider revising the manuscript based on the following comments: 

General Comments:

1. Check the title of this manuscript: “EPDM (ETHYLENE PROPYLENE DIENE)…” Here “MONOMER” is missing.

2. Pay close attention to the language, tone, tense, and format throughout the manuscript. Including but not limited to:

(1) Lines 45-49: Check the grammar of the two sentences, consider to change the full stop to a comma for the below section “…and aging resistance. EPDM has been applied in different fields…”.

(2) Line 148: Check the typo: “...this can (kind?) of materials are called…”; Line 852: “foam” should be capitalized.

(3) Line 149: Check the grammar: “The main feature of these materials is that (they) can be used for…”

(4) Check the grammar of the sentences: Lines 151-152; Line 153; Line 309; Lines 873-874.

(5) Lines 161, 877-881: Check the tense, past tense and present tense are misused.

(6) Lines 178-180, 185: Check spacing between words.

(7) Lines 172, 190, 328, 829: Citations should be put right after the authors of the article mentioned in the manuscript.

(8) Line 637: Check the tone of the sentence, consider revising “in our cars”.

(9) Check the abbreviations through the manuscript. For example, the full name of EPDM appeared multiple times in the manuscript; the full name of TEM appeared for the second time in Line 745; The full name of SEM does not appear when authors first mentioning it. Please also check other abbreviations such as OC, EOC.

(10) Lines 865, 911-912: Check the format of units: cells/mm3, W·m−1·K−1 or W/(m·K).

(11) Lines 378, 503: Check the subscript: t95, t80, Tm.

Introduction:

1. Given that there have been many review articles published in this field, I would suggest acknowledging existing review articles in the field and discussing the emphasis/novelty of the current review.

2. Lines 107-118: Citation(s) should be added.

3. The introduction section is too long, some contents can be moved to other sections, for example, the paragraph starting from Line 119 can be modified and moved to Section 2 Mechanical properties.

All other following sections except conclusions section:

1. Authors walk us through the progress of EPDM by introducing other studies based on different properties. However, Section 7 “EPDM Foams” did not follow this pattern. I would suggest to merge the section 7 into other sections. Alternatively, authors may consider change the format of the manuscript completely – for example, 1. Matrix, 2. Filler dispersing, 3. Filler size, 4. Filler type, 5. Filler materials, etc. and introducing all related properties (mechanical, morphological, Rheological, Swelling) in each section.

2. I recommend to add some subtitles to these sections to improve the readability. For example, in Section 2, subtitles such as 2.1 Filler type; 2.2 Filler size; 2.3 Filler material; 2.3.1 SILP; 2.3.2 Nanoclay.

3. Similar as Table 1, a table is recommended for each section to summarize the related studies.

4. Line 290: The paragraph starting from Line 290, and the followed paragraphs is not related to the mechanical properties, I recommend to add a section named adhesive properties, where authors can discuss the adhesive between fillers and matrix, and between EPDM and other materials.

5. Authors put much effort to review and describe each of the cited articles in detail. I appreciated that, however, I’d expecting more original outputs from the authors, instead of just retell other studies.

6. Line 972: Authors mentioned percolation theory. To make the statement more precise, I recommend that authors should also mention the percolation threshold – the electrical conductivity of the EPDM/nanofiller nanocomposite would not increase significantly only if the addition of fillers reach the threshold.

Conclusions:

This part is too long, a precise and brief paragraph is preferred for this section.

Thank you for considering these comments. I believe that addressing these concerns will strengthen the scientific quality of this manuscript.

Comments on the Quality of English Language

The English language is fine, however, authors should pay attention to the typo, tone, tense, and format throughout the manuscript.

Author Response

Please find attached my response.

Reviewer 2 Report

Comments and Suggestions for Authors

I have carefully reviewed the article, which offers valuable insights into EPDM rubber-based nanocomposites, their performance, and various properties. While the authors have conducted a comprehensive analysis of ongoing research in this field, I have noted several shortcomings in the study. I have attached my questions and suggestions for your consideration.

1.    A graphical abstract by representing a overview of the review article need to be provided.

2.    The classification of EPDM rubbers and types of the modified materials based on it can be shown in flow chart.

3.    Provide a comparison table for reported literatures on EPDM rubber-based nanocomposites, compare their performance and various properties.

4.    Future prospectives of the development of EPDM rubber-based nanocomposites and their possible applications should be discussed in a separate section.

5.    Advantages and disadvantages of EPDM rubber-based nanocomposites and their superiority than other nanocomposites should be discussed.

6.    Sections should be divided into subsections. Like various properties of EPDM rubber-based nanocomposites should be added in a same section followed by the subsections. Various characterizations should be added in the same section. Likewise, the various types of EPDM rubber-based nanocomposites should be added in the same section.

7.     Add a schematic figure for showing the structure of EPDM rubber-based nanocomposite (pictorially). Since this is a review article, addition of more illustrations, figures will make it more attractive.

8.    Do not make frequent paragraph change.

9.    Grammatical mistakes should be checked by using standard software.

10.  I advise the authors to go through the following research articles where various nanocomposite membranes have been made for diverse applications. I wish addition of the important information’s will be helpful to readers: Journal of Hazardous Materials 442 (2023): 129955; Journal of Water Process Engineering, 51 (2023) 103479."

Comments on the Quality of English Language

Need to improvise extensively

Author Response

Please find attached my response.

Reviewer 3 Report

Comments and Suggestions for Authors

This study explores the effects of nanostructures on the mechanical, thermal, and electrical properties of ethylene propylene diene monomer (EPDM), a synthetic rubber. By incorporating nanomaterials, EPDM maintains flexibility while enhancing its tensile and chemical resistance, supporting advancements in synthetic rubber nanocomposites. However, before further consideration of the manuscript, the authors must “fully” address the comments listed below:

1-         Considering the cited improvements in mechanical properties with nanofiller incorporations (ref. Nazir et al.), what are the specific mechanical thresholds (e.g., tensile strength, elongation at break) that differentiate between effective and ineffective levels of nanofiller concentrations for high-voltage insulation applications?

2-         How do varying nanofiller geometries (aspect ratio, specific surface area) impact the rheological properties such as cure rate index and scorch time, especially at different filler concentrations as observed in Khalaf et al.’s study on hybrid EPDM/chitin/nanoclay composites?

3-         With respect to the improved ablation resistance in EPDM nanocomposites mentioned by Guo et al., what specific characteristics of MWCNTs contribute to this phenomenon, and how does the directional effect of the MWCNT network structure influence the in-situ formation of silicon carbide under thermal stress?

4-         Analyzing the work by Lu et al. on highly stretchable sensors, how do varying concentrations of graphene platelets (e.g., 2.9 wt.% vs. 7 wt.%) affect the percolation threshold and thermal conductivity, and what implications does this have for sensor sensitivity and operational stability?

5-         Referring to improvements in char residue and thermal stability in EPDM/Kevlar fiber composites as per George et al., what are the mechanistic interactions between nanosilica and Kevlar fibers that contribute to these enhancements?

6-         What are the specific environmental and economic benefits of using recycled materials (like chitin from shrimp shells) as nanofillers in EPDM, particularly in terms of lifecycle assessment and cost-effectiveness as highlighted in studies by Khalaf et al.?

7-         Investigating the findings from Guo et al., what role do MWCNTs play in influencing the carbothermal reduction reaction in EPDM, and how does this affect the material's suitability for thermal management applications in aerospace engineering?

8-         How does the surface modification of nanofillers impact the interfacial bonding strength and distribution within the EPDM matrix, particularly in relation to the observed effects on mechanical and thermal properties as detailed in the work by Nazir et al. and Guo et al.?

Author Response

Please find attached my response.

Round 2

Reviewer 1 Report

Comments and Suggestions for Authors

Dear Authors,

Thank you for revising the manuscript and the responses. The manuscript has been improved.

I only have some minor comments as below:

1. Lines 126-128: The sentence is italicized, please check.

2. Lines 144, 255, 468, ... : "et al." should be in italic. Please check throughout the manuscript.

3. Lines 1040-1050: While presenting cited contents, authors should use their own words, or paraphrasing. Try to avoid copy and paste the contents from the cited source.

Comments on the Quality of English Language

Authors should check the manuscript carefully, some font issues are spotted.

Author Response

Thank you for revising the manuscript and the responses. The manuscript has been improved.
R: Authors would like to thank the reviewer for his insightful comments and suggestions. The comments were valuable and enhanced the manuscript in many ways.

I only have some minor comments as below:

1. Lines 126-128: The sentence is italicized, please check.
R: Thanks for the notes, and we have made the corrections as suggested.

2. Lines 144, 255, 468, ... : "et al." should be in italic. Please check throughout the manuscript.
R: The text was checked in accordance with the reviewer's suggestion.

3. Lines 1040-1050: While presenting cited contents, authors should use their own words, or paraphrasing. Try to avoid copy and paste the contents from the cited source.
R: We are grateful for the note made by the reviewer, and the text has been rewritten according to your suggestions.

The EPDM rubber-based biocomposites filled with silane-modified wood fiber (SiWF) were foamed using the chemical foaming method by Chen, Gupta, and Mekonnen [180]. Using SiWF and dicumyl peroxide (DCP), the developed EPDM bio-composite foams showed much lower water absorption and improved thermomechanical properties. The tensile strength and modulus of EPDM biocomposite foams exhibited up to 90% and 600% enhancements with SiWF and in situ cross-linking [180]. Figure 9 illustrates the EPDM biocomposite in foam form as shown by the authors [180]. There was a high degree of foaming in the EPDM unfilled. In contrast, biocomposites are less foamed due to the higher concentration of fillers, silane modification, or DCP cross-linking [180].

Reviewer 2 Report

Comments and Suggestions for Authors

Title: EPDM (ETHYLENE PROPYLENE DIENE MONOMER) RUBBER-BASED NANOCOMPOSITES REVIEW: PROPERTIES AND PROGRESS

The authors have revised the text, effectively resolving most of the issues highlighted in the previous review. The updated manuscript now meets the publication criteria; however, it still requires thorough editing, particularly to standardize the reference style. To uphold the journal's uncompromising standards, the authors must restructure their submission. Ensuring the paper meets every criterion is crucial. Ultimately, the editor may decide if the revised manuscript is ready for publication.

Comments on the Quality of English Language

Need to improvise

Author Response

The authors have revised the text, effectively resolving most of the issues highlighted in the previous review. The updated manuscript now meets the publication criteria; however, it still requires thorough editing, particularly to standardize the reference style. To uphold the journal's uncompromising standards, the authors must restructure their submission. Ensuring the paper meets every criterion is crucial. Ultimately, the editor may decide if the revised manuscript is ready for publication.

R: We would like to express our gratitude to the reviewer for the valuable suggestions he provided that helped make the article a more thorough and better article. We agree with all of the points raised and we believe that they will be very useful for future submissions.

Reviewer 3 Report

Comments and Suggestions for Authors

The authors have addressed my comments and the paper can be accepted. 

Author Response

The authors have addressed my comments and the paper can be accepted. 

R: Please accept our sincere gratitude for the valuable suggestions provided by the reviewer which contributed to the improvement of the article.